# Integrating electric field modeling and neuroimaging to explain inter-individual variability of tACS effects

Florian H. Kasten[1,2], Katharina Duecker[1], Marike C. Maack[1], Arnd Meiser[1] & Christoph S. Herrmann[1,2,3]*

Transcranial electrical stimulation (tES) of the brain can have variable effects, plausibly driven by individual differences in neuroanatomy and resulting differences of the electric fields inside the brain. Here, we integrated individual simulations of electric fields during tES with source localization to predict variability of transcranial alternating current stimulation (tACS) aftereffects on α-oscillations. In two experiments, participants received 20-min of either α-tACS (1 mA) or sham stimulation. Magnetoencephalogram (MEG) was recorded for 10-min before and after stimulation. tACS caused a larger power increase in the α-band compared to sham. The variability of this effect was significantly predicted by measures derived from individual electric field modeling. Our results directly link electric field variability to variability of tACS outcomes, underline the importance of individualizing stimulation protocols, and provide a novel approach to analyze tACS effects in terms of dose-response relationships.

[1] Experimental Psychology Lab, Department of Psychology, European Medical School, Cluster for Excellence "Hearing for All", Carl von Ossietzky University, Oldenburg, Germany. [2] Neuroimaging Unit, European Medical School, Carl von Ossietzky University, Oldenburg, Germany. [3] Research Center Neurosensory Science, Carl von Ossietzky University, Oldenburg, Germany. *email: christoph.herrmann@uni-oldenburg.de

Methods to non-invasively modulate brain activity via the transcranial application of magnetic or electrical stimulation are increasingly used in neuroscience to establish causal relationships between specific regions, or activation patterns (e.g., oscillations) in the brain and their behavioral correlates[1,2]. Among these techniques, transcranial electrical stimulation (tES) using weak direct (tDCS) or alternating (tACS) currents are of particular interest as they provide safe and tolerable stimulation at low costs and high portability[3,4]. These features render tES approaches promising for a wide range of clinical applications[5–7]. tDCS is thought to exhibit its effect by changing neuronal excitability via tonic alterations of neuron's resting membrane polarization[1,8–10], whereas the rhythmic shifts in the membrane potentials during tACS are believed to result in neuronal entrainment[2,11,12]. In addition, both methods have been reported to cause changes outlasting the duration of stimulation by several minutes to more than an hour[13–15], likely via NMDA-receptor-mediated plasticity[15–18].

In recent years, tES methods received considerable criticism, arguing that stimulation effects are weak, highly variable, or cannot be replicated[19–22]. Some authors even questioned whether current intensities in the range of 1–2 mA commonly used for tES cause sufficient electric field strengths inside the brain to elicit effects[23,24]. A variety of factors have been identified that can influence effects of non-invasive brain stimulation and may account for its variability[25–30]. A potential major source of tES variability is the influence of individual anatomy and the resulting differences of electric fields inside the brain[31,32]. The development of sophisticated computational models[33–35] allows to study these differences using simulations. Recently, efforts have been carried out to validate the predictions of these models using in vivo electrophysiological recordings in animals and humans[32,36,37]. Results from such simulated electric fields have demonstrated that, when using a fixed stimulation montage and intensity, individual anatomical differences can cause substantial variability of electric fields inside the brain in terms of their spatial distribution and strength[31]. However, if and to which extent these differences explain variability of tES effects on behavioral or physiological outcome measures remains elusive.

In the current study, we investigated whether measures derived from individualized simulations of electric fields and source localization of the target brain activity can be used to explain variability of tACS effects. Specifically, we tested if the spatial correlation of the target brain activity (spatial pattern of the source-projected α-oscillation) with the individually simulated electric field as well as the maximum field strength inside gray and white matter compartments can predict the variability of the power increase in the α-band after tACS. This power increase is relatively well established and has been repeatedly replicated[6,14,17,18,38]. The spatial correlation provides a measure of precision, namely how well the electric field matches the spatial pattern of the targeted brain activity, which is the source of the α-oscillation in the current study. The maximum field strength provides a measure of the intensity at which the target activity can be perturbed. In addition to the spatial precision of the stimulation, the precision of the stimulation frequency has to be considered when targeting brain oscillations using tACS. Recent work emphasized a possible role of the frequency relation between stimulation frequency and the frequency of the target oscillation in the generation of aftereffects[18,39]. Although the frequency of the α-oscillation has long been assumed to be relatively stable, more recent evidence suggests that α-frequency can exhibit substantial intra-individual variability across different tasks and over time[40,41]. To account for this instability, the mismatch between the pre-determined stimulation frequency and the alpha peak frequency before stimulation was included in our analysis. We hypothesized that a model incorporating these factors, which capture the quality of the targeting of stimulation, can explain variability of the power increase in the experimental group after receiving tACS, but not in a control group receiving sham stimulation. Our results indicate that a complex interplay between the spatial precision and strength of the electric field along with the mismatch of the stimulation frequency and participants' individual α-frequency account for a large proportion of the variability of tACS aftereffects in humans.

## Results

**Inter-individual variability of electric fields**. In the first experiment, a total of 40 volunteers received either 20-min of tACS ($n = 20$) or sham ($n = 20$) stimulation at their individual α-frequency (IAF), determined from a short, 3-min resting magnetoencephalogram (MEG) with eyes-open prior to the experiment. Their neuromagnetic activity with eyes-open in rest was recorded for 10-min immediately before and after stimulation (Fig. 1a–c). Based on a structural MRI of each subject, we performed an individual simulation of the expected electric field in the brain. Simulations were used to compute spatial correlations between electric fields and topographies of the α-source (IAF ± 2 Hz) during the pre-stimulation block obtained from a dynamic imaging of coherent sources (DICS) beamformer[42]. In addition, we extracted the average field strength among the 10,000 voxels with the largest electric field values inside gray and white matter compartments as an estimate of the electric field strength reaching the cortex (Fig. 1d).

Although we administered and simulated tACS with a fixed intensity of 1 mA (peak-to-peak) and the same Cz–Oz electrode montage (Fig. 1c), simulations of electric fields revealed differences across subjects in terms of peak electric field magnitude inside the cortex as well as the spatial distribution of electric fields (Fig. 2). On average, electric field strength (average over strongest 10,000 voxels) was $M = 0.13$ V/m $\pm$ SD $= 0.05$ V/m (min $= 0.08$ V/m, max $= 0.36$ V/m). To characterize the similarity of electric fields across subjects, individual simulation results were warped into Montreal Neurological Institute (MNI)-space. Spatial correlations of the fields were computed between all subjects to attain insights into the overall variability of the factor. On average, electric fields correlated with $M = 0.74 \pm$ SD $= 0.05$ ($r_{min} = 0.53$, $r_{max} = 0.85$; Fig. 2 bottom). Spatial correlation between each subjects' α-topography with the simulated electric field was on average $M = 0.55 \pm$ SD $= 0.18$ ($r_{min} = -0.12$, $r_{max} = 0.76$).

**Alpha power increase after tACS**. In accordance with previous findings, a comparison of the source-projected power increase in the α-band from the pre- and post-stimulation blocks of the two experimental groups by means of an independent samples random permutation cluster $t$ test revealed a significantly larger power increase in the tACS group as compared to sham (permutation cluster $t$ test, $p_{cluster} = 0.013$; Fig. 3). No such effect was observed in the β- (permutation cluster $t$ test, all $p_{cluster} > 0.18$) or θ-range (permutation cluster $t$ test, all $p_{cluster} > 0.19$). The two groups did not differ with respect to their source-level α-power during the baseline block (permutation cluster $t$ test, $p_{cluster} = 1$). In both groups, dependent samples cluster permutation $t$ tests against baseline revealed a significant power increase in the α-range from the pre- to the post-stimulation block (permutation cluster $t$ test, tACS: $p_{cluster} < 0.001$; sham: $p_{cluster} = 0.023$; Fig. 4a, f). The power increase in the sham group is limited to few occipital, posterior-parietal and temporal regions, while the power increase in the tACS group spans a wide range of cortical areas covering occipital-parietal, temporal, and frontal areas (Fig. 4a).

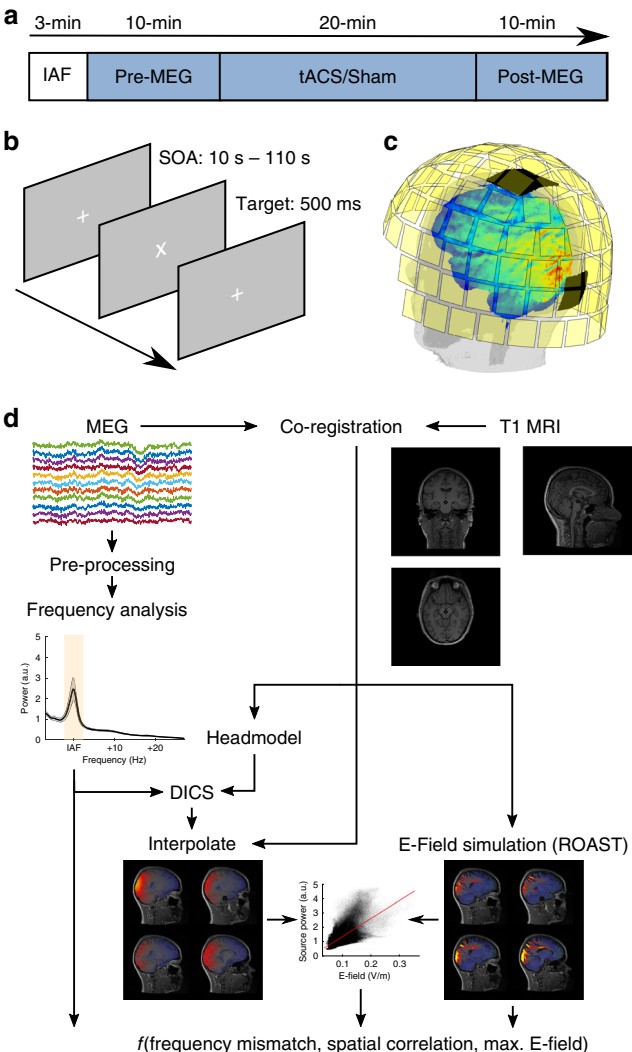

**Fig. 1** Experimental design and analysis pipeline. **a** Time-course of the experiment. Prior to the main experiment, 3-min of eyes-open MEG were acquired to determine participants' individual α-frequency (IAF). After 10-min of baseline measurement, participants received 20-min of tACS at IAF with 1 mA (peak-to-peak) or sham stimulation. Another 10-min of post-stimulation MEG were acquired thereafter. **b** Visual change detection task. Participants were instructed to detect rotations of a white fixation cross, presented on a screen at a distance of ~1 m. **c** MEG sensor array and tACS montage. MEG was acquired from 102 magnetometer and 204 planar gradiometers. Stimulation electrodes were placed centered above positions Cz and Oz of the international 10-10 system. **d** Analysis pipeline to obtain spatial correlation between participants' α-topography and electric field as well as the maximum electric field magnitude inside the gray and white matter compartments and the mismatch between tACS frequency and the dominant frequency during the baseline block.

and individual α-frequency during the baseline block) and STRENGTH (average over 10,000 highest electric field magnitudes inside gray and white matter). This full model was compared with all other possible models with subsets of the above factors using Akaike's Information Criterion (AIC, Supplementary Table 1). The full model was retained for analysis as it exhibited the lowest AIC. Results of the regression analysis indicated that the four predictors explained 76% of the variance (multiple linear model, $R^2 = 0.76$, $F_{15,24} = 5.06$, $p < 0.001$). More specifically, we found that the factor CONDITION (same multiple linear model, $\beta = 2.51e-25$, $t_{24} = 2.38$, $p = 0.03$), as well as interactions between CONDITION*PRECISION$_{Freq}$*STRENGTH (same multiple linear model, $\beta = 2.36e-23$, $t_{24} = 3.06$, $p = 0.005$) and CONDITION*PRECISION$_{Freq}$*PRECISION$_{Spat}$*STRENGTH (same multiple linear model, $\beta = 1.56e-22$, $t_{24} = 3.47$, $p < 0.001$) significantly predicted the power increase. In addition, there was a trend for an interaction of CONDITION*PRECISION$_{Spat}$*PRECISION$_{Freq}$ (same multiple linear model, $\beta = 2.36e-24$, $t_{24} = 2.05$, $p = 0.052$).

All significant predictors explaining participants' power increase involved the factor CONDITION. This pattern of results was expected given that our predictors are intended to relate to the efficacy of tACS and should thus not be suited to explain variance in the sham group. To specifically test that this is the case, we separately fitted a model with factors PRECISION$_{Spat}$, PRECISION$_{Freq}$ and STRENGTH to the data of the two experimental groups. In the tACS group, the model significantly predicts participants' power increase (multiple linear model, $R^2 = 0.87$, $F_{7,12} = 11.5$, $p < 0.001$; Fig. 4b–e, Fig. 5a). The factors PRECISION$_{Spat}$ (same multiple linear model, $\beta = 1.68e-24$, $t_{12} = 4.26$, $p = 0.001$) and PRECISION$_{Freq}$ (same multiple linear model, $\beta = 1.27e-25$, $t_{12} = 2.41$, $p = 0.003$), as well as interactions of PRECISION$_{Spat}$* STRENGTH (same multiple linear model, $\beta = 2.64e-23$, $t_{12} = 2.99$, $p = 0.01$), PRECISION$_{Spat}$*PRECISION$_{Freq}$ (same multiple linear model, $\beta = 2.62e-24$, $t_{12} = 4.29$, $p = 0.001$), PRECISION$_{Freq}$* STRENGTH (same multiple linear model, $\beta = 2.03e-23$, $t_{12} = 5.79$, $p < 0.001$) and PRECISION$_{Spat}$* PRECISION$_{Freq}$* STRENGTH (same multiple linear model, $\beta = 1.53e-22$, $t_{12} = 6.18$, $p < 0.001$) significantly predicted participants' power increase. Again, AIC suggests that this full model is superior to all other possible models with fewer predictors (Supplementary Table 2). As expected, the model fails to predict the power increase in the sham group (multiple linear model, $R^2 = 0.13$, $F_{7,12} = 0.26$, $p = 0.96$; Fig. 4g–j, Fig. 5b). In line with these results, the lowest AIC was obtained for an intercept model omitting all predictors related to stimulation, further confirming that the model is not suited to explain data of the sham group (Supplementary Table 3). Two alternative versions of the analysis predicting only the maximum power increase within each of the group ROIs, and correlating each participant's alpha power increase in a certain voxel with the electric field strength in that same voxel are presented in Supplementary Note 1 and Supplementary Figure 2.

**E-Field variability predicts power increase after tACS.** To evaluate whether the observed inter-individual differences of electric fields account for the variability of our outcome measure, for each subject, the average power increase between pre- and post-stimulation was extracted from the two group specific clusters, that is the cluster of each group exhibiting significant power increase from the pre- to the post-stimulation block (Fig. 4a, f). The results were submitted to a multiple linear regression analysis with factors CONDITION (tACS vs. sham), PRECISION$_{spat}$ (spatial correlation of α-topography with electric field), PRECISION$_{Freq}$ (mismatch between stimulation frequency

**Peripheral model fails to predict power increase after tACS.** Recently, concerns have been raised that tACS may not cause its effects by direct electric stimulation of the brain, but rather indirectly via stimulation of peripheral nerves (e.g., stimulation of the retina or transcutaneous nerves)[43,44]. Our results indicate that the extent of the tACS aftereffect can be predicted using the electric field inside the brain, which is difficult to explain with such peripheral mechanisms of action. We therefore conducted an additional analysis aiming to explain the data in our tACS group by a model incorporating the maximum current in the skin (STRENGTH$_{skin}$; average over the maximum 10,000 voxels

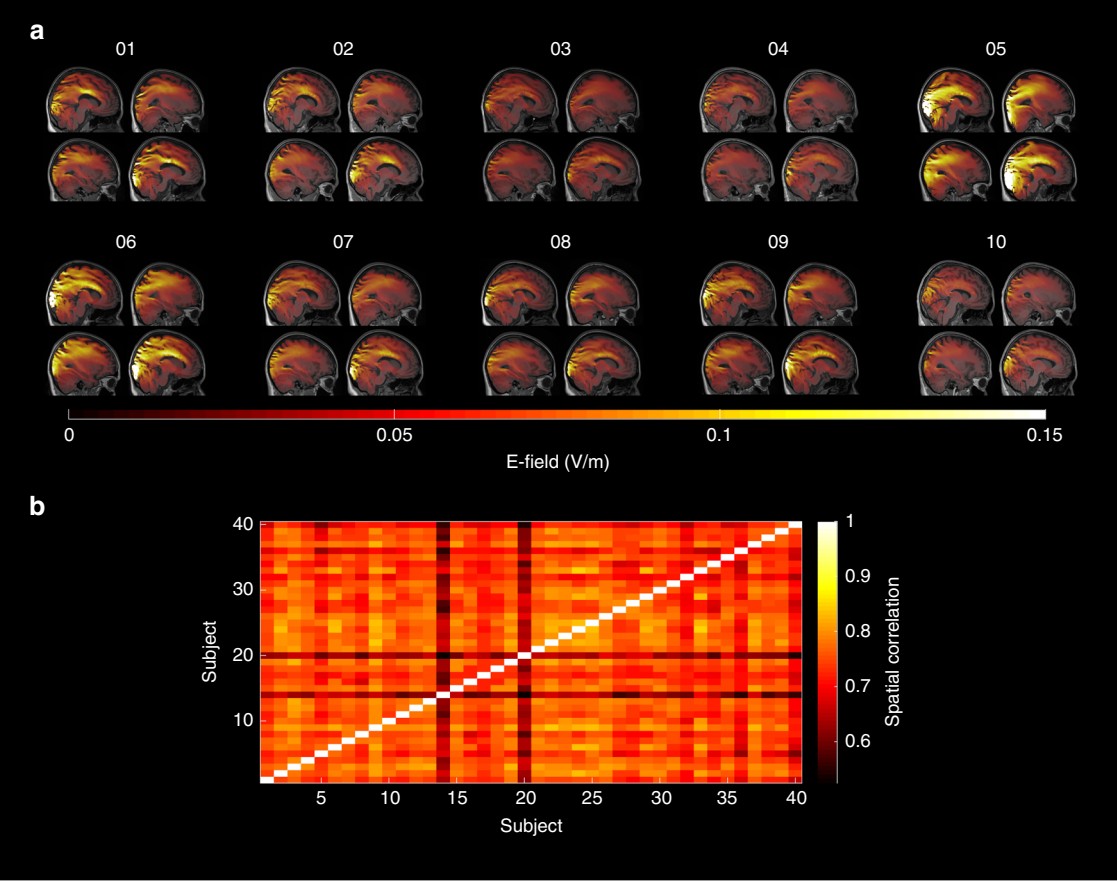

**Fig. 2** Variability of electric fields across subjects. **a** Simulations of the electric fields inside the brain resulting from the Cz–Oz configuration applied at 1 mA (peak-to-peak) exemplified for the first ten subjects. Simulations were performed on the individual brain and warped into Montreal Neurological Institute (MNI) space for visualization purposes. Overall simulation results show a quite large variability between subjects. Please refer to Supplementary Figure 1 for an overview of all simulations in the sample **b** Spatial correlations of electric fields between all subjects in MNI-space.

within the skin compartments) and the eyeballs (STRENGTH$_{eye}$; average over the maximum 1000 voxels within the eyeballs). In addition, we included the factor PRECISION$_{Freq}$ from our initial model as a similar effect of frequency mismatch has to be expected for peripheral stimulation effects. The resulting model was not able to significantly predict the power increase after tACS (multiple linear model, $R^2 = 0.22$, $F_{7,12} = 0.49$, $p = 0.82$). Based on AIC, no possible model incorporating a subset of these factors was superior to a simple intercept model in explaining the data (Supplementary Table 4). More importantly, none of the models was superior to the previous model incorporating the electric field in the brain (Supplementary Table 3,4).

**Model validation and replication.** Because the model explains a striking amount of variance in the tACS group (~ 87%), we performed a leave one out cross-validation (LOOCV) to obtain a more conservative estimate of the explained variance. LOOCV can be used to perform cross-validation on small datasets. The model is trained (fitted) $n$ times on $n-1$ datapoints and then used to predict the response variable for the remaining data point. This way, we can estimate how well the model generalizes to new observations. Based on the predictions of the LOOCV (Fig. 5c), we recomputed $R^2$. Results suggest that the model still explains more than half (51.5%) of the variance in the tACS group ($R^2 = 0.52$). Noteworthy, the tACS group contains two subjects that apparently exhibited exceptionally strong alpha power increases relative to baseline (Fig. 3b), both of which were very well

predicted by the cross-validation model (Fig. 5c). Even when the model was trained without these two subjects, it predicted the comparatively strong power increase based on their electric field parameters and frequency mismatches (Supplementary Note 2, Supplementary Figure 3).

In order to investigate how much of participants' individually sham-controlled tACS effect the model can explain, and in order to replicate the previous results, we repeated the experiment using a within-subject design on a sample of 19 subjects. On two separate days, participants received tACS or sham stimulation for 20-min. The order of tACS and sham conditions was counter-balanced across participants.

Similar to the first experiment, a dependent samples random permutation cluster $t$ test revealed a stronger increase of participants' source-projected α-power after tACS as compared to sham (permutation cluster $t$ test, $p_{cluster} < 0.001$, Fig. 6a-d). Source-projected α-power significantly increased after tACS (permutation cluster $t$ test, $p_{cluster} < 0.001$), and showed a trend towards increased α-power after sham stimulation ($p_{cluster} = 0.08$). Again, there was no significant difference in the neighboring β- (permutation cluster $t$ test, $p_{cluster} > 0.16$) and θ-frequency range (permutation cluster $t$ test, $p_{cluster} > 0.08$). Subsequently, we tested whether participants' individual stimulation effects in the α-band (power increase after tACS −power increase after sham) within the significant cluster (Fig. 6a) are explained by our measures of tACS targeting. To this end, the power increase relative to sham was submitted to a multiple linear regression with factors PRECISION$_{Spat}$, PRECISION$_{Freq}$ and STRENGTH. Although only reaching trend-level, the model again

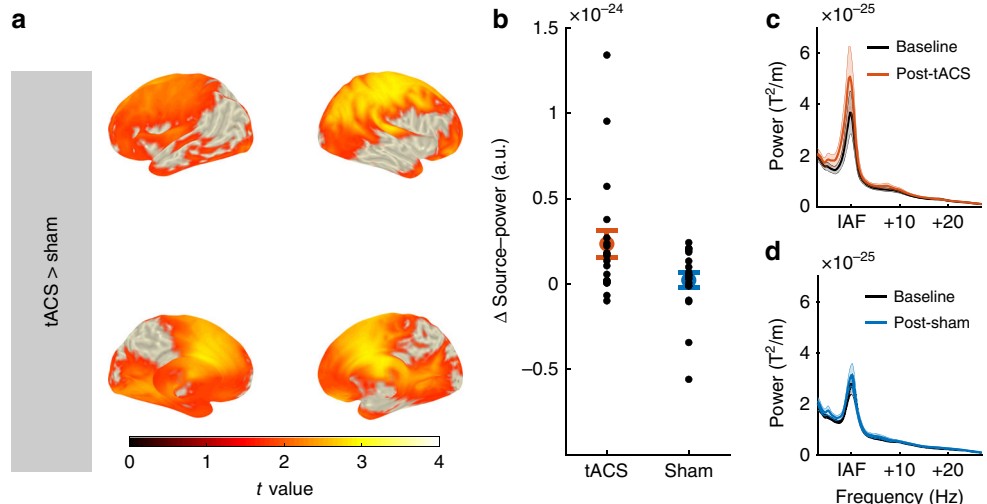

**Fig. 3** Effect of tACS on source-level α-power. **a** Statistical map contrasting the power increase from the pre- to the post-stimulation block between experimental conditions (tACS vs. sham). Statistical map shows $t$ values, thresholded at an α-level of 0.05. **b** Power increase within the cluster for each of the experimental groups. Black dots represent the power increase of each individual subjects in the experimental groups. **c** Power spectra before and after tACS (average over all gradiometer sensors and participants). All spectra were aligned to participants' individual α-frequency (IAF) before averaging. **d** Power spectra before and after sham stimulation. Error bars and shaded areas depict standard error of the mean (S.E.M.). Source data are provided as a Source Data file.

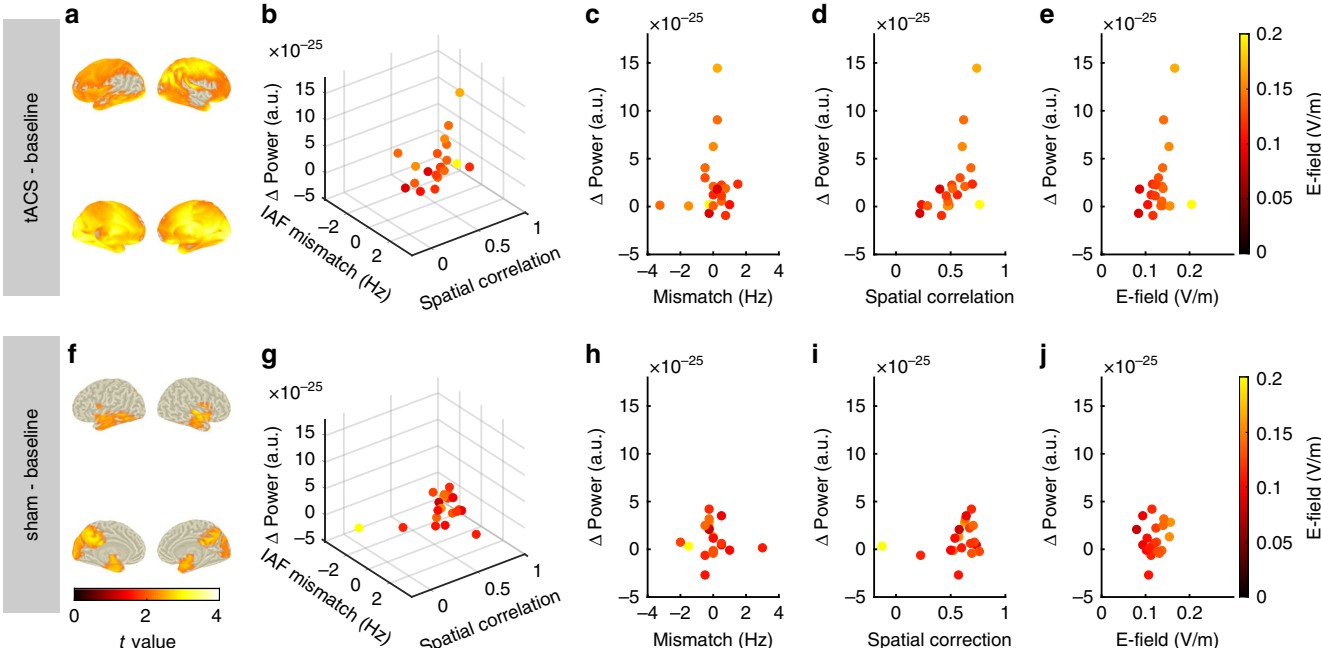

**Fig. 4** Power increase as a function of spatial correlation, field strength, and frequency mismatch. **a** Statistical map contrasting post-tACS vs. pre-tACS power in the α-band. Statistical map depicts $t$ values thresholded at an α-level of 0.05. The cluster was used as ROI to extract the individual power increase of each subject in the tACS group for the subsequent regression analysis **b–e**. **b** Power increase of the tACS group as a function of frequency mismatch between tACS frequency and the dominant frequency during baseline, the spatial correlation between the simulated electric field and the source-level α-topography during baseline, and the maximum field strength in gray and white matter compartments. Each dot represents data of a single subject. **c–e** Same data as in **b** shown for each predictor of the model. **f** Statistical map contrasting post-sham vs. pre-sham power in the α-band. Statistical map is thresholded at an α-level of 0.05. The cluster was used as ROI to extract the individual power increase of each subject in the sham group for the subsequent regression analysis **g**. **g** Power increase of the sham group as a function of frequency mismatch between tACS frequency and the dominant frequency during baseline, the spatial correlation between the simulated electric field and the source-level α-topography during baseline, and the maximum field strength in gray and white matter compartments. Each dot represents data of a single subject. **h–j** Same data as in **g** shown for each predictor of the model. Source data are provided as a Source Data file.

explains a striking amount of the variance of the tACS effect (multiple linear model, $R^2 = 0.62$, $F_{7,11} = 2.66$, $p = 0.07$; Fig. 5d, Fig. 6e–h). Specifically, PRECISION$_{Spat}$ significantly predicted participants' individual stimulation effect (same multiple linear model, $\beta = 1.073e{-}24$, $t_{11} = 3.07$, $p = 0.01$). In addition, there

was a trend towards an interaction between PRECISION$_{Spat}$, PRECISION$_{Freq}$, and STRENGTH (same multiple linear model, $\beta = -3.6e{-}23$, $t_{11} = -1.75$, $p = 0.1$). Although the model performs weaker on the replication data set, results are generally in agreement with the findings of the previous experiment.

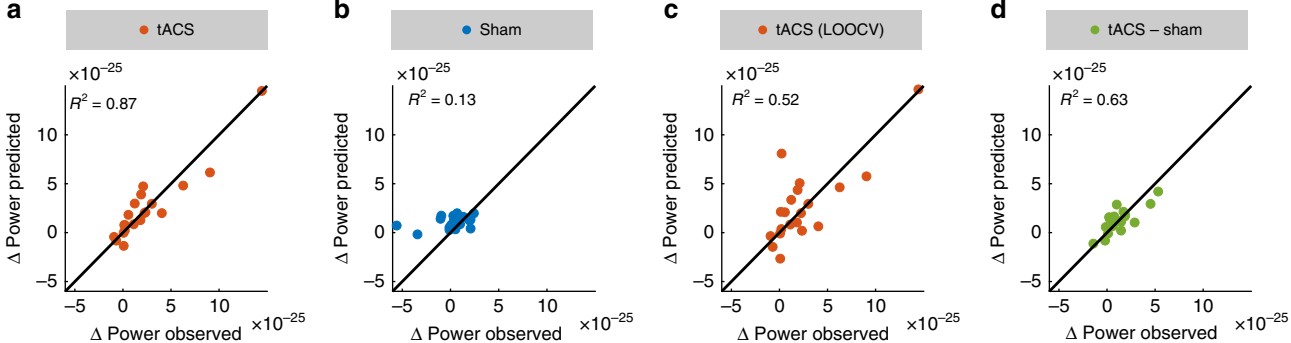

**Fig. 5** Model predictions. Scatterplots depict the power increase from the baseline to the post-stimulation block predicted by the statistical models, plotted against empirically observed values. Each dot represents data of a single subject. The diagonals indicate the line of perfect prediction (predicted Δpower = observed Δpower). The distance of each data point from the diagonal indicates the residual prediction error. **a** Predicted and empirical power increase for the tACS group. **b** Predicted and empirical power increase in the sham group. **c** Power increase predicted for each subject in the tACS group by the cross-validated model. For each data point $n_i$ the model was trained on $n − 1$ observations to predict the remaining $i$th observation. **d** Individually sham-controlled tACS effect (power increase relative to baseline after tACS−power increase relative to baseline after sham) on source-level α-power predicted by measures of tACS targeting. Source data are provided as a Source Data file.

## Discussion

Increasing the reliability of low-intensity tES is one of the major challenges for the brain stimulation community. An understanding of the factors determining successful modulation of outcome measures by tES is crucial as the field is advancing these techniques towards clinical applications[5–7]. In the current study, we demonstrated that the variability of tACS aftereffects can be explained by an interplay of factors qualitatively capturing the targeting of the stimulation.

In line with previous findings[31], our simulations indicate that electric fields induced at a fixed intensity with the same electrode montage vary quite substantially on an individual level. We were able to directly link this variability to the outcome of tACS. When integrated with neuroimaging, simulations of electric fields can be used to derive qualitative measures of the targeting, i.e., the spatial correlation between electric field and target, the maximum electric field strength inside the brain and the mismatch between stimulation frequency and frequency of the target oscillation. Together, these measures explained a substantial proportion of variance (∼ 51–87%) of our outcome measure (power increase in the α-band after tACS). In contrast to this, the model did not explain any variance of the outcome measure after sham stimulation. In our second experiment, measures of tACS targeting explained ∼ 63% of the variability of the individually sham-controlled stimulation effect in the sample. These results emphasize the importance of individualizing stimulation parameters for example by taking individual anatomy and the resulting electric field differences into account. Advancing algorithms for electric field modeling toward individualized electrode montages maximizing the field strength at the desired target[45], and closed-loop stimulation systems adapting stimulation parameters to the current brain activity[46] may greatly improve reliability of brain stimulation effects. This is especially important in clinical settings where the reliability of stimulation determines whether or not a patient's symptoms improve.

In the context of research applications, study designs may benefit from incorporating individualized electric field modeling and neuroimaging for statistical analysis. We believe that this approach has some advantages over the pure comparison of group means, which is commonly used to investigate stimulation effects. Such comparisons implicitly assume that tES exerts consistent effects across participants. Especially, when using "one-fits-all" stimulation protocols, a high prevalence of non-optimal targeting and the resulting numbers of potential low- or non-

responders may compromise the sensitivity of such statistical approaches to detect stimulation effects. In contrast, the statistical model proposed here tests for stimulation effects by assessing whether the variability of the outcome measure follows a dose–response relationship that would be expected based on the proposed underlying mechanisms. Consequently, the model is not only robust against low- and non-responders, but rather expects low- or non-responsiveness in cases where the standard stimulation protocol does not fit the individual subject well. As a further advantage, this mechanistic modeling largely rules out alternative explanations of the observed effects. In the field of tACS, concerns have been raised that stimulation effects could be explained by peripheral effects such as visual entrainment owing to phosphenes[43] (a perception of flickering lights resulting from a polarization of the retina[47]) or transcutaneous stimulation of peripheral nerves[44]. Although our experiment did not rule out a possible impact of peripheral stimulation by applying scalp anesthetics or a focused stimulation montage, the aftereffect observed in our study seems very unlikely to be explained by predictors derived from the electric field inside the brain if such peripheral mechanisms had primarily caused the effect. This was supported by our alternative model incorporating the electric fields in the skin and the eyeballs failing to predict participants' power increase after tACS. To the contrary, as the model links the stimulation effect to variations of electric fields inside the brain results provide supporting evidence that tACS applied in the range of 1 mA can be sufficiently strong to elicit aftereffects arising from polarization of brain tissue. However, the impact of stimulation seems to depend on the strength and precision of the individual electric field and the precision of the stimulation frequency.

When the strength of tES-induced electric fields necessary to modulate neuronal activity is discussed, electric fields reaching the human brain are usually compared against thresholds derived from animal studies[11,48–50]. Those thresholds are in the range of 0.2 V/m to 0.5 V/m[48]. While evidence from animal models can strongly contribute to our understanding of the underlying mechanisms of tES methods, there are crucial discrepancies between experimental designs in animal and human studies that may limit the translation of voltage thresholds. In animal models, stimulation is usually applied to in vitro brain slices or to localized neural assemblies via intracranial stimulation electrodes in vivo. The modulation of neuronal activity is measured during short trains of stimulation, in the range of few seconds. In human

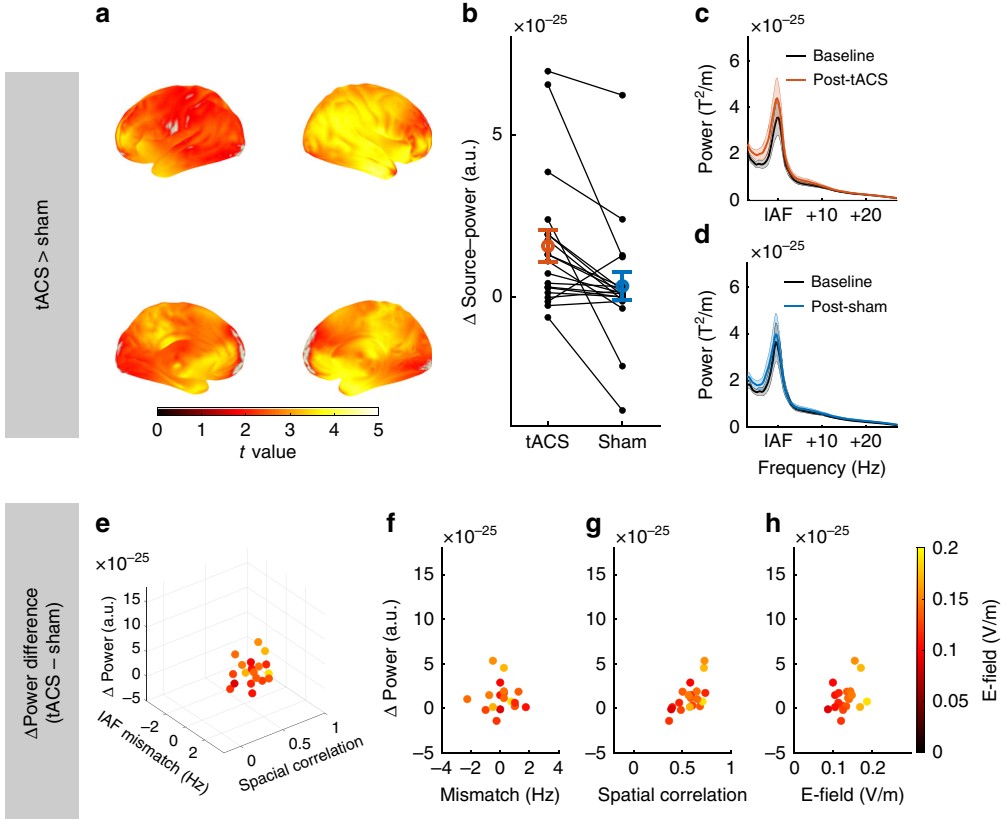

**Fig. 6** Results of within-subject replication experiment. **a** Statistical map contrasting the power increase from the pre- to the post-stimulation block between experimental conditions (tACS vs. sham) in the α-band. Statistical map shows $t$ values, thresholded at an α-level of 0.05. **b** Power increase within the cluster for each of the experimental conditions. Black dots/lines represent the power increase of each individual subjects in the experimental conditions. **c** Power spectra before and after tACS (average over all gradiometer sensors and participants). All spectra were aligned to participants' individual α-frequency (IAF) before averaging. **d** Power spectra before and after sham stimulation. **e** Individually sham-controlled tACS effect (power increase relative to baseline after tACS−power increase relative to baseline after sham) as a function of function of frequency mismatch between tACS frequency and the dominant frequency during baseline, the spatial correlation between the simulated electric field and the source-level α-topography during baseline, and the maximum field strength in gray and white matter compartments. Each dot represents data of a single subject. **f–h** Same data as in **e** shown for each predictor of the regression model. Error bars and shaded areas depict standard error of the mean (S.E.M.). Source data are provided as a Source Data file.

studies, however, tES protocols commonly feature stimulation durations of several minutes (often > 10-min), with stimulation applied to comparably large areas of the brain. Consequently, stimulation effects may build up over longer periods of time or amplify via large-scale neuronal interactions[51]. Individual simulations suggest electric fields in our study were in the range of 0.1 V/m − 0.2 V/m, providing evidence that the electric field strength necessary to elicit stimulation effects in humans may be in the lower range of those thresholds derived from animal models (or even below). More research will be necessary to determine the electric field strength required to modulate neuronal activity in humans (e.g., by testing stimulation protocols comparable to human experiments in animal models) and allow more informed discussions about tACS efficacy.

Despite tuning tACS to each participants' individual α-frequency as measured prior to the experiment, there was still a mismatch between the stimulation frequency and the individual α-frequency observed during the experiment that significantly contributed to the variability of the power increase after tACS. This mismatch has previously been reported to occur despite applying stimulation at participants' individual frequency and to affect the extend of tACS aftereffects[18,39]. Different processes may explain the occurrence of a frequency mismatch between tACS and brain oscillations. First, the dominant frequency in a specific band may underly changes over time. For example, systematic

drifts of the individual α-frequency have been observed over time and depending on the background task[40,41]. Second, for practical reasons in the current study the identified power peak in the α-band was rounded to the next integer frequency naturally giving rise to mismatches between stimulation frequency and the frequency of the targeted brain oscillation. Given the impact of this factor, future studies might benefit from improved procedures to estimate tACS frequency.

Besides the investigation into the role of individual electric fields for tACS effects, the current study is among the first to perform source localization of the tACS-induced power increase in the α-band. Although the effect has been repeatedly replicated, results usually rely on data from few electrode sites, providing little information about its spatial extend[14,17,18,38]. To our surprise, the effect of tACS in the α-band was very widespread covering a large proportion of the cortex, including frontal areas not covered by our electrode montage. We did not further investigate this observation up to this point as it was beyond the scope of our main research question. However, there is evidence that distributed brain networks communicate via correlated activity within specific frequency bands[52]. It might thus also be possible that the tACS-induced modulation of oscillatory activity within a circumscribed region could lead to co-stimulation of distant brain areas functionally coupled via the stimulated frequency band. It should, however, also be emphasized that

differences in cluster extent are not independent of oscillatory power and might thus be solely explained by the power enhancement in the α-band. In addition to the power increase after tACS, we also observed an increase of power in the α-band after sham stimulation that was not explained by our statistical model. Such increase in α-power over time is commonly observed with time-on-task and has been associated with vigilance decrement and mental fatigue[40,53–56].

As with all scientific studies, some limitations of the current findings deserve consideration. The individual electric fields used for our analysis were obtained from computational modeling. This approach can only provide predictions of the individual electric field with an inherent degree of uncertainty and simplification. For example, errors in the automatic tissue segmentation can add random error to the estimated field strengths. Recently, first efforts have been carried out to validate and calibrate results of current flow predictions using in vivo electrophysiology[32,36,37]. Results of these studies suggest that the models perform very well in predicting the spatial distribution of the induced electric fields, whereas tending to overestimate their strength. For our analysis approach, an accurate prediction of the exact field strength is not necessarily required, as long as the relative difference in the fields across subjects is accurately represented and uncertainties in the estimates, e.g., due to segmentation errors, introduce error variance but no systematic bias. Noteworthy, conductivities used for simulations of the ROAST toolbox have recently been calibrated to increase the accuracy of voltage and field strength predictions[36]. Our results indicate that both, the spatial distribution and the field strength predicted by individual electric field models, contain meaningful information allowing to predict the impact of tACS aftereffects, indicating that the computational models are sufficiently accurate to capture inter-individual differences. Nevertheless, further validation and optimization of electric field modeling using empirical data will be necessary to increase confidence in their predictions. Especially when models are integrated in the analysis of physiological or behavioral outcome measures as we propose in the current study, the accuracy of the utilized computational model will be crucial. This also holds true if the models are used to rule out peripheral stimulation effects. In the current study, we tested whether the current inside the skin or the eyeballs might be better suited to explain the observed power increase, which seemed not to be the case. This can be seen as evidence against an influence of such mechanisms on our effects. However, it is possible that the utilized computational models do not optimally predict the electric fields inside these compartments. To increase confidence in this kind of control analysis, future studies could assess whether the electric field modeling is able to accurately predict stimulation effects if peripheral mechanisms are explicitly targeted (e.g., by a montage optimized to stimulate the retina rather than the brain). Evidence for reliable separation of stimulation effects originating inside the brain and in the periphery would support the potential of electric field model to substitute additional control conditions currently required to rule out peripheral mechanisms to be underlying tACS effects.

The predictions of the tACS aftereffect in the current study are based on predictors derived from the magnitude of the electric field inside the brain, ignoring the direction of the field relative to the cortical surface. Although the current flow radial to the cortical surface (or normal component of the electric field) determines the strength of somatic polarization of cortical pyramidal cells, the current flow radial to the cortical surface polarizes horizontally arranged cortico-cortical axons[57]. In principle, these different components of the electric field may differentially contribute to stimulation effects. Models could incorporate this contribution by computing predictor variables, i.e., spatial correlations with the brain activity of interest and the strength of the electric field, separately for each of the field directions. In the current experiment, we refrained from applying such models to the data as we aimed to keep statistical models sufficiently simple and interpretable.

Another important aspect to be discussed is the generalizability of the current results. Together with the mismatch of the stimulation frequency, individual differences of the electric fields explained a striking amount of the power increase in the α-band after tACS, pointing toward the significance of individual anatomy and the resulting differences in electric fields for tACS effects and potentially tES effects in general. Although the proposed underlying mechanisms of the different tES approaches differ[1,2], they all ground on the principle that the electric fields induced to the brain alter the resting potential of neurons[8,11], with stronger electric fields at the target area causing larger polarity changes. As this fundamental dose–response relationship is captured by our statistical model, it seems likely that individual differences in electric fields may have a similar impact for other tES methods or outcome measures. In the current study, we focused on the development of an analysis pipeline to investigate the impact of electric field differences on tES outcomes and tested it on a well replicated effect. Further work is needed to determine the exact impact of these differences for the various types of tES methods (tDCS, tACS, tRNS, etc.) and physiological and behavioral outcome measures, as well as for on- and offline effects. With the current work, we provide a powerful analysis framework, adaptable to EEG-source localization or fMRI that can strengthen our understanding of the contribution of individual anatomy on tES outcomes and the mechanisms of tES in general.

## Methods

**Participants**. In experiment one, 40 healthy volunteers (age: $24 \pm 3$ years, 20 females, 20 males) without history of neurological or psychiatric disease were randomly assigned to one out of two experimental groups (tACS or sham) in a single-blind design. Groups were counterbalanced for participants' sex. In experiment two, 22 healthy volunteers received tACS or sham stimulation on one out of two experimental sessions on separate days. The order of stimulation conditions was counterbalanced across participants. Both experimental sessions took place at the same time of day and were spaced at least 4 days apart. One participant aborted the experiment after the first session. Two participants indicated extreme levels of tiredness after the experiment as well as too short sleep durations the night prior to the experiment and were excluded from the study. Thus, 19 participants (11 females, age: $25 \pm 3$ years) remained for analysis. Two subjects participated in both experiments.

All subjects were right-handed according to the Edinburgh Handedness-Scale[58] and had normal or corrected to normal vision. All were non-smokers and reported to be medication-free at the day of the measurement. Subjects gave written informed consent prior to the experiment and received monetary compensation for participation. Both experiments were approved by the Commission for Research Impact assessment and Ethics at the University of Oldenburg and performed in accordance with the declaration of Helsinki.

**Magnetoencephalogram**. Neuromagnetic signals were acquired at a rate of 1 kHz using a 306-channel whole-head MEG system with 102 magnetometer and 204 orthogonal planar gradiometer sensors (Elekta Neuromag Triux System, Elekta Oy, Helsinki, Finland), housed in an electrically and magnetically shielded room (MSR; Vacuumschmelze, Hanau, Germany). Five head position indicator (HPI) coils were attached to participants' head prior to the recording. Their positions were digitized along with the location of three anatomical landmarks (nasion, left and right tragus) and >200 head-shape samples using a Polhemus Fastrak (Polhemus, Colchester, VT, USA). Participants were seated underneath the sensor array in upright position (60° dewar orientation). To determine participants' IAF, a 3-minute recording of spontaneous MEG activity with eyes-open was acquired before the main experiment. Signals were filtered between 1 Hz and 40 Hz and segmented into 2-sec epochs. Fast Fourier Transforms (FFT; Hanning window) were computed for each of the segments and the resulting spectra were averaged. The power peak in the α-band between 8 Hz and 12 Hz within a fixed set of posterior gradiometer sensors (a detailed list is provided in the Supplementary Methods) was identified and the closest integer frequency to the identified peak was used as stimulation frequency during the following experiment. MEG was recorded with continuous head position tracking during two experimental blocks, one pre- and one post-stimulation (Fig. 1a). Although the recording was continued during

stimulation, signals acquired during this period were discarded from the analysis owing to the massive electromagnetic stimulation artifact, that can currently not be reliably removed from resting state recordings[59–63].

**Electrical stimulation**. Electrical stimulation was administered via two surface conductive rubber electrodes positioned centered over locations Cz (7 × 5 cm) and Oz (4 × 4 cm) of the international 10-10 system (Fig. 1c). Electrodes were attached to participants' scalp using an electrically conductive, adhesive paste (ten20 paste, Weaver & Co, Aurora, CO, USA). The sinusoidal stimulation waveform was digitally sampled in Matlab 2016a at a rate of 10 kHz and streamed to a digital analog converter (Ni-USB 6251, National Instruments, Austin, TX, USA) connected to the remote input of a constant current stimulator (DC Stimulator Plus, Neuroconn, Illmenau, Germany). The stimulator was placed in an electrically shielded cabinet outside the MSR. From there, the signal was gated into the MSR via a tube in the wall using the MRI extension-kit of the stimulator (Neuroconn, Illmenau, Germany). Electrode impedance was kept below 20 kΩ (including two 5 kΩ resistors in the stimulator cables). Prior to the experiment, participants were introduced to potential visual and somatosensory sensations during stimulation and subsequently familiarized with the stimulation by brief application of tACS at the frequency and intensity used during the main experiment. Following a 10-min baseline period, participants received either 20-min of tACS at IAF or sham stimulation. Stimulation was applied with an intensity of 1 mA (peak-to-peak) and two 10-sec fade-in/fade-out intervals at the beginning and end of the stimulation period, respectively. During sham stimulation, tACS was applied during the first 30-sec of the stimulation period (including 10-sec fade-in and fade-out). All other stimulation parameters were kept similar. Stimulation frequency was on average $M = 10.1 \text{ Hz} \pm SD = 1 \text{ Hz}$ ($M_{tACS} = 9.9 \text{ Hz} \pm 1 \text{ Hz}$; $M_{Sham} = 10.4 \text{ Hz} \pm 0.6 \text{ Hz}$) for the first experiment and 10.5 Hz ± 1.1 Hz ($M_{tACS} = 10.5 \text{ Hz} \pm 1.1 \text{ Hz}$; $M_{Sham} = 10.5 \text{ Hz} \pm 1.1 \text{ Hz}$) for the second experiment.

After the recordings, participants filled out a questionnaire assessing common adverse effects of tES[64] and indicated whether they believe they received tACS or sham stimulation. Subsequently, participants were informed about their true experimental condition and the goals of the study. Results of the debriefing are presented in Supplementary Notes 3 and 4.

**Vigilance task**. To ensure that participants remained awake and attentive during the 40-min measurement, they performed a visual change detection task similar to previous studies[14,17,39,65] (Fig. 1b). Visual stimuli were presented using Matlab 2016a, using Psychtoolbox 3[66]. Stimuli were rear-projected (Panasonic PT-DS 12 KE, 60 Hz refresh rate) onto a screen inside the MSR at a distance of ~ 100 cm. At the center of the screen a white fixation cross extending 0.45° visual angle) was presented on a gray background. The fixation cross was rotated by 45° for a duration of 500 ms at random intervals with an SOA of 10–110-sec (Fig. 1b). Participants were asked to react to the rotation by pressing a button with their right index finger.

**MRI acquisition**. To perform source analysis and individual simulations of electric fields during stimulation, a structural MRI was obtained from each subject. Images were acquired using a Siemens Magnetom Prisma 3 T whole-body MRI machine (Siemens, Erlangen, Germany). A T1-weighted 3-D sequence (MPRAGE, TR = 2000 ms, TE = 2.07 ms) with a slice thickness of 0.75 mm was used.

**Data analysis**. Software: Data analysis was performed in Matlab 2016a (The MathWorks, Inc. Natick, MA, USA) using the Fieldtrip toolbox[67] for MEG data processing and ROAST v. 2.7[35] for individualized electric field modeling. Statistical analysis of source-level data was performed using statistical functions provided by the Fieldtrip toolbox. All other statistical analyses were performed using R 3.5.1 (The R Core Team, R Foundation for Statistical Computing, Vienna, Austria).

MEG preprocessing: External interference in the MEG was suppressed using the spatiotemporal signal space separation method (tSSS), with standard settings ($L_{in} = 8$, $L_{out} = 3$, correlation limit = 0.98)[68,69] using MaxFilter v2.2 (Elekta Neuromag, Elekta Oy, Finland). Movement compensation was performed using the continuous HPI signals[70]. The tSSS method decomposes the MEG signal into spatiotemporal components originating from inside and outside the sensor helmet. The method is commonly used to suppress external artifacts and interference signals, especially those originating in the proximity of the head (e.g., implants, deep-brain stimulator, etc.)[68,69,71]. The method is thus well suited to remove interference brought into the MEG helmet via the cables connected to the stimulation electrodes. Further, it allows to compensate for head-movements by transforming the signals to the initial head position[70]. Signals were subsequently imported to Matlab and resampled to 250 Hz. A 4th-order forward-backward Butterworth filter introducing zero phase-shift between 1 Hz and 40 Hz was applied. Artifacts reflecting heart-beat, eye-movements or muscle activity were manually removed using independent component analysis. After visual inspection of component topographies and time-courses an average of 3.7 ( ± SD: 1; min: 3, max: 8) components were removed before back-projecting the signals into sensor-space. In experiment 2, 3.6 ( ± SD: 0.9; min: 2, max: 6) components were rejected on average. Rejection criteria were based on recommendations in the literature[72]. Signals were cut into 2-sec epochs. Segments still containing artifacts were rejected.

FFTs (4-sec zero-padding, Hanning window) were computed on each of the segments. The resulting power spectra were averaged across the first 260 artifact-free segments in each experimental block.

DICS beamforming: Power in the individual α-band (IAF ± 2 Hz) was projected into source-space using a DICS beamformer[42] utilizing all 306 (magnetometer and gradiometer) channels. A common spatial filter was computed from the averaged cross-spectrum in the IAF band across all segments of the two experimental blocks. Data were projected onto an equally spaced 6 mm grid, warped into MNI (Montreal Neurologic Institute) space. Single-shell head-models[73], derived from individual T1-weighted MRIs, co-registered to participants' head position inside the MEG were used. Regularization was set to $λ = 1e-12$. The common filters were then applied to project data of the pre- and post-stimulation block. For each source location, the power difference between the pre- and post-stimulation block was computed. To test whether the power increase in the α-band was larger after tACS as compared to sham, power differences were submitted to a one-sided non-parametric random permutation cluster $t$ test with 10,000 randomizations and Monte Carlo estimates to calculate $p$ values. The approach allows to test for statistical differences in large-scale data sets without the need for prior assumptions about the location of effects and while controlling for multiple comparisons. We decided to test for stimulation effects exclusively in the source-space in order to circumvent additional variance owing to individual differences in head-size or position inside the MEG helmet, which naturally limit the analysis of MEG sensor-space signals. In addition, power in the α-band before and after stimulation was compared separately for both groups using random permutation cluster $t$ tests for dependent samples. The identified clusters were used as region of interests (ROI) to extract the average power increase from the pre- to the post-stimulation block for subsequent analysis (see next section). To evaluate frequency specificity of the effects, the analysis was repeated for the individual theta (IAF – 5 Hz to – 3 Hz) and beta (IAF + 4 to + 20 Hz) band.

Individualized electric field calculations: Individual simulations of the electric field induced by the Cz – Oz montage were performed on the co-registered, T1-weighted MRI of each subject using the ROAST toolbox v2.7[35]. The toolbox offers some advantages over other modeling tools currently available, as it requires comparably short computation times (~ 25-min per subject), automatically determines standard EEG electrode positions in individual head-space and provides results in Matlab format, allowing easy integration with source-level MEG results. As part of the ROAST pipeline, a 6-compartment (white matter, gray matter, csf, bone, skin, air), finite-element model is created from individual MRIs using the SPM12 segmentation algorithm. A post-processing routine is subsequently used to optimize the segmentation output for electric field modeling (for details, see[35]). Simulations were run with an injected current of 0.5 mA (corresponding to 1 mA peak-to-peak), a 7 × 5 cm rectangular electrode patch over electrode site Cz, and a 4 × 4 cm rectangular electrode patch over electrode site Oz. Electrodes were modeled with a thickness of 2 mm and a 2 mm layer of gel. Default conductivities of the toolbox were used for the different compartments. Recently, these have been validated/calibrated based on intracranial recordings of 10 human epilepsy patients[36].

To capture the inter-individual variability of the electric fields across subjects, we computed two measures, one indicating the spatial precision of stimulation (PRECISION_{Spat}: how well does the electric field overlap with the targeted brain activity), the other indicating the strength of stimulation inside the brain (STRENGTH). As a measure of precision, we calculated the spatial correlation between the electric field with the individual topography of each participant's α-power. To this end, participants' IAF band power (IAF ± 2 Hz) during the pre-stimulation block was localized using a DICS beamformer. Data were projected onto an equally spaced 3-mm grid defined in individual head-space (without warping onto a standard brain). Filters were computed using the cross-spectra in the IAF band obtained from the artifact-free segments of the baseline block. To account for the center-of-head bias of the beamformer, the neural activity index (NAI) was computed. The NAI is the source activation at each dipole location normalized by an estimate of the noise at that location. NAIs were subsequently interpolated onto the individual, T1-weighted MRI, which has the same resolution as the electric field calculation and thus allowed us to compute the spatial correlation between the source-projected α-topography and the individual electric field profile for each subject. To index the STRENGTH of stimulation inside the brain, we identified the 10,000 voxels inside the gray and white matter compartments of each subjects' simulation result showing highest electric field magnitude and computed the average electric field magnitude across these voxels.

To evaluate whether these measures of electric field differences account for the variability of our outcome measure, we modeled each subject's power increase within the ROIs as a function of CONDITION (tACS vs. sham), PRECISION_{Spat}, STRENGTH and PRECISION_{Freq} with a multiple linear regression model. PRECISION_{Freq} captures the mismatch between the pre-determined stimulation frequency (sf) and the dominant frequency (df) observed during the baseline block (mismatch = sf – df), extracted from the average spectrum over all sensors. An overview of each subject's individual alpha frequency, stimulation frequency and their mismatch are provided in Supplementary Table 5 and 6.

**Reporting summary**. Further information on research design is available in the Nature Research Reporting Summary linked to this Article.

## Data availability

The data that support the findings of this study are available upon reasonable request from the corresponding author C.S.H. The data are not publicly available owing to potentially identifying information that could compromise participant privacy. The source data underlying Fig. 2 (bottom), Fig. 3b–d, Figs. 4,5, Fig. 6b–h and Supplementary Figure 3 are provided as a Source Data file.

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

## Acknowledgements
We thank Helge Ahrens, Dr. Tina Schmitt, Gülsen Yanc, and Katharina Grote for the acquisition of structural MRIs and for supporting the MEG data collection. This research was supported by the Neuroimaging Unit of the Carl von Ossietzky University Oldenburg funded by grants of from the German Research Foundation (3 T MRI INST 184/152-1 FUGG and MEG INST 184/148-1 FUGG). Christoph S. Herrmann was supported by a grant of the German Research Foundation (SPP, 1665 HE 3353/8–2).

## Author contributions
F.H.K. and C.S.H. conceived the study; F.H.K. and K.D. programmed the experiment; F.H.K., K.D., and A.M. acquired data for experiment 1, F.H.K., and M.C.M. acquired data for experiment 2, F.H.K. analyzed the data, all authors wrote the manuscript.

## Competing interests
C.S.H. holds a patent on brain stimulation and received honoraria as editor from Elsevier Publishers, Amsterdam. The remaining authors declare no competing interests.
