## [Peer Review File · Nature Communications]

Reviewers' comments:

Reviewer #1 (Remarks to the Author):

In the present work, Kasten and co-workers investigated whether the variability observed in the effects of alpha-band transcranial alternating current stimulation (tACS) can be accounted by inter-individual differences in estimated electric fields, obtained by means of magnetoencephalography and magnetic resonance imaging. The present findings point towards an individualized setting of stimulation parameters by taking into account individual anatomy and resulting electric fields differences.

The study is technically of interest and addresses a critical issue in the literature on tACS and tES in general, providing a contribution to existing knowledge of non-invasive brain stimulation community.

Below I raise a few points which need clarification.

Title

Since the study specifically tested the effects of tACS in the alpha band, and the results as they are cannot be directly generalized to other tES methods, I think that the title may be more representative of the content of the study by replacing "tES" with "tACS" (but see the below comment).

Nevertheless, I am not totally convinced about the claims and results, it seems that authors had a specific aim in using ~10 Hz (i.e., IAF) tACS: emphasis is given in the introduction about the stimulation frequency and reliability of previous results on tACS and alpha power. Nonetheless present results are reported as "generic" and therefore the final claim seems a post doc interpretation. So, I'm wondering if the main result can be considered that tACS is equivalent to e.g., tDCS and therefore the frequency of the delivered oscillations is not so relevant. The control group in this experiment was a sham group while a different transcranial electric stimulation type would have been more appropriate, e.g., why not testing theta or beta or tRNS? Why not testing the proposed model changing stimulation parameters base on model results? These control conditions would render the results more remarkable.

Methods and Results

1) The authors described a mixed-design study, and tested the effect of time (pre vs. post stimulation) in the tACS and in the sham groups separately. I think it would be also important to know whether the two groups statistically differ at baseline, as Figure 3c may suggest (e.g., is the alpha power level before tACS comparable between groups? Does it account for variability in tACS outcome?). Controlling for baseline activity is important because the functional state of participants at the time of stimulation represents another possible candidate to account for tES variability.

2) After stimulation, the authors reported a significant increase in alpha power in the tACS as well as in the sham group. Although the tested factors did not account for variability in the sham group, I think that the increase in alpha power after sham should be further discussed.

3) The MEG system used in the experiment provides both magnetometers and gradiometers, and for sake of completeness it may be clarified if the analyses were performed on both sensor types, and, eventually, why data from one sensor type have been discarded.

4) Since information about participants' guess on the type of stimulation (i.e., sham vs real) was collected by the experimenters, I was wondering if the authors may report more information about it (e.g., was there a difference between the two groups?). To this regard, I was also wondering

whether all participants were naïve to tES-induced sensations (e.g., if not, how many of them were naïve? Was this factor counterbalanced between groups?).

Others

- Line 102: for clarity, please specify that whether the increase in power was in the alpha band.
- Line 111: the cortical areas showing the alpha power increase in the tACS group may be described in greater detail.
- Line 292: the authors reported that the power peak in the alpha band was identified within “posterior gradiometer sensors”; were the same sensors across all participants? How were they selected?
- Figure 1a: it seems that “MEG” is missing when indicating the “Post- “ MEG measurement in the timeline of the experiment.

Reviewer #2 (Remarks to the Author):

The manuscript proposes a new modeling/statistical approach that aims to predict how much a certain subject's MEG alpha power will increase in response to tACS. The model uses individual MRI scans to estimate the tACS generated electric field distribution and strength for each subject. In addition, the frequency mismatch between tACS and the individual alpha frequency is included in the regression model, as is the overlap between a subject's MEG alpha topography and the tACS generated E-field.

The often weak and variable responses to tACS (and the related methodology tDCS) are a major concern for the field. This manuscript does go some way to proposing potential causes for this variability – namely individual head anatomy leading to 1) different electric field strengths and 2) different spatial distributions of the electric field and 3) frequency mismatch between tACS and the targeted oscillation.

However, the experimental design, data collection and data analysis suffer from a number of major concerns, which I explain in detail below. The proposed approach and experimental data also completely ignore the largest issue facing the tACS field at the moment – namely are tACS effects caused by the electric field in the brain (which may be too weak to have a significant neuromodulatory effect) or by stimulation of cranial nerves such as the optic nerve (phosphenes) or nerves giving input to the somatosensory system. Since the data collection methods did not directly address these major concerns, it's very difficult to know exactly what type of stimulation was causing the reported effect on MEG alpha power. Was the effect driven by stimulation of cranial nerves or the electric field in the brain? Without an experimental approach or data collection method that address these fundamental concerns, it's difficult to judge the validity of the proposed statistical modeling approach to predict effect size.

Major Concerns

1. As stated above the first major concern relates to what is causing the observed increase in MEG alpha – the field in the brain or stimulation of peripheral nerves? The authors should use an experimental design that allows for the separation of these two potential causes, e.g. focused montage and scalp anesthetic.
2. The effect size of tACS on MEG alpha power appears to be very small and to be driven by just two subjects. See Fig. 3b and Fig. 4c, d and e. Without the two subjects showing alpha power increases close to 1 and $1.5e-24$, the sham and the tACS groups would be unlikely to show any significant difference. Are these two subjects outliers?
3. In general results appear to be cloaked behind complex processing and statistics. This makes it difficult to judge the quality of the data and the size of the effect. The main effect of tACS on MEG alpha power is reported in section 2.2, where the authors use a random permutation cluster t-test

on the source-projected alpha power. Why was this test used? The authors should report the results of a simple, standard, non-parametric test, such as Wilcoxon sum-rank and signed-rank test, on the MEG alpha power with appropriate simple filtering (is source-projection necessary to detect an effect?). It is also not clear why MEG alpha power increase in both the tACS and sham group.

The same applies for the use of clustering analysis. Why are only alpha power changes within specific clusters compared? What would happen to the effect if alpha power changes from the whole brain were included? The clusters used in the sham and tACS groups appear to be very different. Wouldn't a fairer comparison be to take the same brain regions in each group? What happens to the effect on alpha power if this approach is used? It's often unclear how and where the clustering analysis was used. Specifically in lines 113 to 120, alpha power increase was extracted from specific clusters, were the PRECISIONfreq, PRECISIONspat and STRENGTH extracted from the same clusters?

4. If the authors are confident in their model's prediction power it should be properly validated with a second set of independent experiments. Take a new group of subjects, now use the model designed from the first set of experiments to predict how much each new subject's alpha power will increase. Then measure the actual alpha power change for each new subject and show that the model (based only on data from the first group) correctly predicts alpha power changes.

Further critical validations that could easily be carried out are missing:

a) If the tACS frequency – alpha frequency mismatch is important, then simply systematically change this parameter in an experimental setting for a number of subjects and show a relationship between frequency mismatch and alpha power.

b) If E-field amplitudes is important, then systematically change this parameter (stimulation at 0.5, 1 and 1.5 mA) and show a relationship between E-field amplitude and alpha power.

c) If overlap with the alpha topography is important, then change electrode configurations and show a clear change in alpha power.

Without these extra validations, the model/statistical prediction as presented is not completely convincing.

5. The authors used ROAST for the segmentations of individual MRI data. ROAST is known to perform sub-optimally (compared to manual segmentation) in CSF and bone segmentation [1]. Both CSF and bone thickness will significantly contribute to the electric field strength and distribution in the brain. Therefore, given that we know this type of automated segmentation will introduce variability in individual head models, it is somewhat strange and puzzling that the authors find these head models can explain variability in the experimental data. This should be explained.

[1] Huang, Y., Datta, A., Bikson, M. & Parra, L. C. Realistic volumetric-Approach to Simulate Transcranial Electric Stimulation – ROAST – a fully automated open-source pipeline. bioRxiv 217331 (2019). doi:10.1101/217331

Minor Comments

There appears to be a word missing from the first sentence in the abstract. I would suggest that 'reducing variability' or 'understanding variability' is the challenge. Every measurement or intervention has variability.

Introduction, line 47. 'fail to replicate' something missing in this sentence

Line 62-63 'the spatial correlation of the target brain activity (source of the α -oscillation) with the individually simulated electric field' It's not clear what this means

Line 66-67 'The former can be interpreted as a measure of precision (how well does the electric field match the pattern of the targeted brain activity),' It's not clear what this means. Which pattern is being referred to? Anatomy? Neural oscillation? It's also not completely clear what the 'former' and 'latter' refer to here.

Line 74-75. The Introduction could be rounded off with a short statement of how the data/results support (or refute) the hypothesis stated in the last sentence.

Figure 1. What does 'activation' refer to on the correlation plot at the bottom of the figure? Units?

Line 90-91. 'inter individual differences across subjects' Should be either 'inter individual differences' or 'differences across subjects'

Figure 2. It's good to show all the individual data. But maybe some could be moved to supplemental and the authors could pick out a few examples to show on a larger scale. MNI abbreviation in the figure legend is not clear.

Line 97. MNI abbreviation is not spelled out when it is first used.

Line 97-100. It's not clear (at least at this point in reading the manuscript) why looking at the spatial correlation between different individuals is interesting or relevant.

Line 104. sample's -> samples

Figure 3. Why is the power shown in arbitrary units?

(Line 118, Line 132-133) If this is the prediction, then why include the sham group in the model? Only model tACS group?

Line 116 – not clear what these clusters are?

Reviewer #3 (Remarks to the Author):

Kasten et al., tested whether the inter-individual variability in the after-effects of 1 mA tACS in Oz-Cz montage at individual alpha frequency (IAF) on spontaneous posterior alpha power (i.e., a power increase in the alpha band) can be explained by several factors regarding the individual spatial and frequency specificity of the stimulation: i.e., the overlap of individual cerebral stimulation current distribution (estimated from anatomically informed electric field simulations) and topography of the targeted alpha oscillatory power (determined from resting state MEG source localization), the maximal current strength in the brain (from the simulations), and the divergence of the stimulation frequency (determined from a 3 min resting MEG session prior to the experiment) from the actual individual alpha frequency (determined during an MEG baseline block). They compared their experimental group (N = 20) to a control group (N = 20) that received conventional sham stimulation only in a between subject-design. The authors report remarkably high R^2 coefficients of 87 % (original full model) to 52 % (leave one out cross-validation) explained variance.

This is a highly interesting paper. It is clearly written and the data processing and statistical analyses are sound. The conclusions are principally backed up by the results, although there are some limitations (see below). If these exciting results hold true, this paper would set the ground for establishing individualized stimulation protocols that allow the induction of after-effects (and

presumably also immediate online effects, such as neuronal entrainment) with conventional low currents (here 1 mA peak-to-peak). This is in line with several recent studies showing considerable effects of low-intensity tACS on cognition (such as working memory or memory encoding) when precisely targeting networks in an individually tailored fashion. This paper is also timely and of particular importance given recent studies questioning the effectiveness of low-intensity tACS altogether. However, given the relevance and potential implications of the reported findings, I would encourage the authors to further corroborate their conclusions by implementing at least some of the control experiments suggested below. I have the following specific comments:

1. The decision for a between-subject design over a within-subject design is unfortunate (in particular because of the huge inter-individual variability which is in the focus of this paper), as it prevents the prediction of individual sham-controlled tACS after-effects. Instead the authors had to employ a general linear model including the stimulation (tACS vs sham) as factor and the spatial and frequency specificity as covariates (or a multiple linear regression analysis, as the authors put it, with the stimulation as between-subject dummy variable, and interactions with that factor as relevant comparisons. Luckily, however, the predictors also explained substantial variance for the tACS group alone but not for the sham group, however in that case obviously without a direct statistical tACS-sham comparison and with a reduced sample size of 20 subjects, which is at the edge for a proper correlation analysis. Also, the leave-one-out cross-correlation is not a very strong validation check, and yet it was associated with a considerable drop from 87 % to 53 % explained variance. It would therefore be much more convincing if the results would be replicated with a new independent sample, preferably in a within-group design.

2. The most convincing evidence would be, however, if the authors would specifically optimize the current flow in a given subject (based on its electric field simulations) using a multi-electrode setup for targeting its individual alpha power distribution and compare it to the standard Cz-Oz montage. If the optimized setup produces much larger after-effects, this would be a very strong argument.

3. As the authors discuss, recent evidence published in this journal (Asamoah et al., 2019) has demonstrated the impact of transcutaneous stimulation effects, and also retinal stimulation has been highlighted as potential confound (Schutter, 2016). The authors argument against such an influence explaining their effects is reasonable, i.e., that the predictive value of the overlap between current distribution and target oscillation topography in the brain cannot be explained easily with such sensory confounds. However, the predictive value of the current density in the skin and eye balls has not been investigated, and there is no other control condition, such as a different stimulation frequency or different electrode positions. Is the observed alpha power increase thus really due to synaptic changes following alpha entrainment? Or would another frequency or stimulation target have resulted in similar predictions? Maybe the spatial correlation between current distribution and alpha power distribution is mainly driven by individual differences in the alpha power topography? A frequency control (stimulating at flanker frequencies) or montage control (stimulating in a different electrode configuration) may thus strengthen the conclusions.

4. Page 6: The authors tested for frequency specificity only with respect to the EEG power spectrum, and did not find neighboring theta and beta range to be affected. However, with $p = 0.18$ and $p = 0.19$, the beta error (false negative) due to low power is presumably not very low... and may have been statistically significant with a larger N?

5. Page 6: comparing cluster extent after DICS beamforming is problematic, even after a significant cluster permutation test, since differences in cluster extent are not independent from differences in oscillatory power. Is there really a broader activation or just a stronger activation of the same network (and would lowering the threshold reveal the same regions in the sham condition)?

6. Also: Is there a direct link between the voxel-wise stimulation current strength and the subsequent increase in alpha power in that voxel?

7. Caption Figure 4: "The cluster from a group analysis was used to extract individual alpha power increase values of each subject in the tACS group for subsequent regression analysis" Using a group ROI for extracting that information introduces a bias in a way that it may result in larger values for subjects with an alpha topography more similar to the group average (and thus with a larger overlap of their individual cluster and the group average cluster). The approach is understandable. But what happens if using the x largest values from voxels of the individual alpha power cluster for extraction? The same holds true for extracting the maximum current strength from the simulations. Why not try to extract it only for the relevant voxels, i.e., the ones overlapping with the (individual?) alpha power cluster?

8. It is surprising that the deviation of the tACS frequency from the IAF was a relevant predictor, given that tACS frequency had deliberately been adjusted to the IAF beforehand. How can there still be a meaningful divergence? Is this explained by a substantial change in IAF from resting MEG to baseline MEG session? What is the actual resolution of calculating the IAF for each measurement and what is the resolution of the actual stimulation frequency (1 Hz, 0.5 Hz, 0.25 Hz?). The actual IAF values and stimulation frequency values should be reported in detail and the issue should be discussed.

9. Only current strength but not current direction relative to the cortical surface is investigated. This factor should at least be discussed.

10. Page 3: "While tDCS is thought to exhibit its effect by changing neuronal excitability via alterations of neuron's resting membrane polarization, tACS is believed to work via the principles of neuronal entrainment." How is this entrainment effect mediated? By rhythmic shifts in the membrane potential. The immediate mechanisms of action on the neuronal level should be the same for tDCS and tACS, but the consequences may be different.

11. Please elaborate on the tSSS method. (page 18)

12. Page 19: please provide criteria for ICA component rejection as well as mean and SD for the rejected components.

References

Asamoah B, Khatoun A, Mc Laughlin M (2019) tACS motor system effects can be caused by transcutaneous stimulation of peripheral nerves. *Nat Commun* 10:266.

Schutter DJ (2016) Cutaneous retinal activation and neural entrainment in transcranial alternating current stimulation: A systematic review. *Neuroimage* 140:83-88.

Response to reviews

Reviewers' comments:

Reviewer #1 (Remarks to the Author):

In the present work, Kasten and co-workers investigated whether the variability observed in the effects of alpha-band transcranial alternating current stimulation (tACS) can be accounted by inter-individual differences in estimated electric fields, obtained by means of magnetoencephalography and magnetic resonance imaging. The present findings point towards an individualized setting of stimulation parameters by taking into account individual anatomy and resulting electric fields differences.

The study is technically of interest and addresses a critical issue in the literature on tACS and tES in general, providing a contribution to existing knowledge of non-invasive brain stimulation community.

Below I raise a few points which need clarification.

Title

Since the study specifically tested the effects of tACS in the alpha band, and the results as they are cannot be directly generalized to other tES methods, I think that the title may be more representative of the content of the study by replacing "tES" with "tACS" (but see the below comment).

We thank the reviewer for pointing at this important aspect. Based on the suggestion we have changed the title to: "Integrating electric field modelling and neuroimaging to explain inter-individual variability of tACS aftereffects"

Nevertheless, I am not totally convinced about the claims and results, it seems that authors had a specific aim in using ~10 Hz (i.e., IAF) tACS: emphasis is given in the introduction about the stimulation frequency and reliability of previous results on tACS and alpha power. Nonetheless present results are reported as "generic" and therefore the final claim seems a post doc interpretation. So, I'm wondering if the main result can be considered that tACS is equivalent to e.g., tDCS and therefore the frequency of the delivered oscillations is not so relevant. The control group in this experiment was a sham group while a different transcranial electric stimulation type would have been more appropriate, e.g., why not testing theta or beta or tRNS? Why not testing the proposed model changing stimulation parameters base on model results? These control conditions would render the results more remarkable.

We thank the reviewer for raising these important points. Our goal in the study was to establish a link between the variability of electric fields and stimulation effects, specifically those of tACS (as this is the technique our research group is mostly working with). One major problem with establishing such a link is where to start as the family of tES techniques offers an excessive parameter space to choose from (stimulation method, frequency, location...).

We decided to target alpha oscillations with ~10 Hz tACS because it is the best replicated effect in the tACS literature which comes in handy for the interpretation of the data. If the model would have failed to predict tACS effects in a not well-established stimulation protocol it would have been difficult to decide whether the electric field modelling is not predictive of the effect, or if there is no effect of the chosen protocol in the first place.

The rationale behind using a sham control was to rule out that our model predicts participants' power increase based on some general relationship between the alpha source topography/ the brain's conductivity or alike, but specifically predicts the effect of tACS. Therefore, it was thus important to us to demonstrate that the model fails to predict the power in an experimental group that did not receive any stimulation.

It was also not our intention to overstate the generalizability of the findings, but rather to point out that since all tES methods ground on the same fundamental principle of changing the resting potential of neurons, the complex dose-response relationship that we find for tACS might also hold for other techniques (omitting the influence of the frequency mismatch which is specific for tACS of course). This does, however, not imply that all these methods (tDCS, tACS, tRNS) are equivalent and cause the same effects. During tDCS, a constant polarization of neural tissue is applied that causes increased/decreased likelihoods of neurons to fire, while during tACS phases of de- and hyperpolarization of the cell membranes occur in a rhythmic fashion which is thought to give rise to neural entrainment. This has of course to be tested. However, our experimental design was specifically tailored to tACS and is not well suited to evaluate other tES methods, because effects of these methods on brain oscillations are not as well established as for tACS.

Finally, we agree with the reviewer that a model-based, individual optimization of stimulation parameters would be remarkable. However, the implementation of such individually tailored stimulation is unfortunately not easy to achieve. Our model is not able to suggest stimulation montages based on individual source localization results. Implementing an algorithm for this purpose will need intensive software development and testing. In order to achieve accurate stimulation frequencies a closed-loop stimulation system incorporating online tACS artifact correction would be necessary which is not feasible at this point.

In order to support our results, we decided to focus on a replication of the current results in a within subject design, allowing us to reveal the explained variance of the individually sham controlled tACS effect. We found that our model explains a large proportion of variance of participants' individual tACS effect in the alpha band relative to sham.

Methods and Results

1) The authors described a mixed-design study, and tested the effect of time (pre vs. post stimulation) in the tACS and in the sham groups separately. I think it would be also important to know whether the two groups statistically differ at baseline, as Figure 3c may suggest (e.g., is the alpha power level before tACS comparable between groups? Does it account for variability in tACS outcome?). Controlling for baseline activity is important because the functional state of participants at the time of stimulation represents another possible

candidate to account for tES variability.

We thank the reviewer for bringing up this point. We ran an additional analysis testing for source-level differences in alpha power during the baseline period. The result of this analysis did not reveal any significant differences between groups ($p = 1$). In order to keep our statistical models as simple as possible, we thus refrained from including the factor to our analysis.

Ln 114 “The two groups did not differ with respect to their source-level α -power during the baseline block ($p_{cluster} = 1$).”

2) After stimulation, the authors reported a significant increase in alpha power in the tACS as well as in the sham group. Although the tested factors did not account for variability in the sham group, I think that the increase in alpha power after sham should be further discussed.

We thank the reviewer for emphasizing this important aspect. In fact, increase in alpha power over time (e.g. with time on task) is commonly observed in EEG and MEG. We have added a reference to existing literature reporting such effects to the discussion section:

Ln 301: “In addition to the widespread power increase after tACS, we also observed an increase of power in the alpha band after sham stimulation that was not explained by our statistical model. Such increase in alpha power is commonly observed with time-on-task and has been associated with vigilance decrement and mental fatigue^{39,50–54}”

3) The MEG system used in the experiment provides both magnetometers and gradiometers, and for sake of completeness it may be clarified if the analyses were performed on both sensor types, and, eventually, why data from one sensor type have been discarded.

We thank the authors for this suggestion. We clarified the sensors utilized for performing the analysis in the methods section.

Ln 460: “Power in the individual α -band (IAF \pm 2Hz) was projected into source-space using a DICS (dynamic imaging of coherent sources) beamformer³⁹ utilizing all 306 (magnetometer and gradiometer) channels.”

In addition, we are now listing the channels used to determine participants' individual alpha frequency prior to the experiment in the Supplementary Materials.

4) Since information about participants' guess on the type of stimulation (i.e., sham vs real) was collected by the experimenters, I was wondering if the authors may report more information about it (e.g., was there a difference between the two groups?). To this regard, I was also wondering whether all participants were naïve to tES-induced sensations (e.g., if not, how many of them were naïve? Was this factor counterbalanced between groups?).

We thank the reviewer for this suggestion. Prior to the experiment, all subjects were informed about possible sensations of the stimulation and briefly familiarized with the stimulation to-be applied during the experiment. Consequently, they were not naïve

towards the to be expected sensations. We have added a short description of the procedure to the methods section.

Ln: 403: “Prior to the experiment, participants were introduced to potential sensations (visual and somatosensory) and subsequently familiarized with the stimulation by brief application of tACS at the frequency and intensity used during the main experiment.”

Further we added results of the analysis of participants’ guesses as Supplementary Materials.

Others

- *Line 102: for clarity, please specify that whether the increase in power was in the alpha band.*

We have added the requested information to the results section (now in Ln 110).

- *Line 111: the cortical areas showing the alpha power increase in the tACS group may be described in greater detail.*

We have added more detailed descriptions about the regions in Ln 118.

- *Line 292: the authors reported that the power peak in the alpha band was identified within “posterior gradiometer sensors”; were the same sensors across all participants? How were they selected?*

We thank the reviewer for this suggestion. We have added a list with the channels used for the IAF determination to the Supplementary Materials.

- *Figure 1a: it seems that “MEG” is missing when indicating the “Post- “ MEG measurement in the timeline of the experiment.*

We thank the reviewer for spotting this error. We have corrected the figure.

Reviewer #2 (Remarks to the Author):

The manuscript proposes a new modeling/statistical approach that aims to predict how much a certain subject’s MEG alpha power will increase in response to tACS. The model uses individual MRI scans to estimate the tACS generated electric field distribution and strength for each subject. In addition, the frequency mismatch between tACS and the individual alpha frequency is included in the regression model, as is the overlap between a subject’s MEG alpha topography and the tACS generated E-field.

The often weak and variable responses to tACS (and the related methodology tDCS) are a major concern for the field. This manuscript does go some way to proposing potential causes for this variability – namely individual head anatomy leading to 1) different electric field strengths and 2) different spatial distributions of the electric field and 3) frequency mismatch between tACS and the targeted oscillation.

However, the experimental design, data collection and data analysis suffer from a number of major concerns, which I explain in detail below. The proposed approach and experimental data also completely ignore the largest issue facing the tACS field at the moment – namely are tACS effects caused by the electric field in the brain (which may be too weak to have a significant neuromodulatory effect) or by stimulation of cranial nerves such as the optic nerve (phosphenes) or nerves giving input to the somatosensory system. Since the data collection methods did not directly address these major concerns, it's very difficult to know exactly what type of stimulation was causing the reported effect on MEG alpha power. Was the effect driven by stimulation of cranial nerves or the electric field in the brain? Without an experimental approach or data collection method that address these fundamental concerns, it's difficult to judge the validity of the proposed statistical modeling approach to predict effect size.

Major Concerns

1. As stated above the first major concern relates to what is causing the observed increase in MEG alpha – the field in the brain or stimulation of peripheral nerves? The authors should use an experimental design that allows for the separation of these two potential causes, e.g. focused montage and scalp anesthetic.

We thank the reviewer for raising this very important point. From Ln 239 in the manuscript we are discussing the concerns in the field about the potential contribution of peripheral mechanisms such as stimulation of cutaneous nerves or visual entrainment via polarization of the retina (phosphenes). We argue that our result that the power increase after tACS can be predicted by the electric field inside the brain cannot be easily explained by peripheral mechanisms. In other words, if peripheral mechanisms had caused the effect, the electric field inside the brain should not have predicted the tACS effect. However, we agree with the reviewer that this deserves further testing. We thus created an alternative statistical model aiming to predict the power increase after tACS using the maximum current inside the skin compartment and inside the eyeballs. This alternative model did not explain the power increase after tACS ($R^2 = .22$, $F_{7,12} = 0.49$, $p = .82$) and was not superior to the model incorporating electric field metrics based on AIC (Supplementary Table S3, S4). The additional analyses are now part of the results section:

Ln 158: “Recently, concerns have been raised that tACS effects may not originate from electric stimulation of the brain, but exhibit its effects indirectly via stimulation of peripheral nerves (e.g. stimulation of the retina or transcutaneous nerves)^{42,43}. Our results indicate that the extent of the tACS aftereffect can be predicted using the electric field inside the brain, which is difficult to explain with such peripheral mechanisms of action. We therefore conducted an additional analysis aiming to explain the data in our tACS group by a model incorporating the maximum current in the skin (*STRENGTH_{skin}*; average over the maximum 10,000 voxels within the skin compartments) and the eyeballs (*STRENGTH_{eye}*; average over the maximum 1000 voxels within the eyeballs). In addition, we included the factor *PRECISION_{Freq}* from our initial model as a similar effect of frequency mismatch has to be expected for peripheral stimulation effects. The resulting model was not able to significantly predict the power increase after tACS ($R^2 = .22$, $F_{7,12} = 0.49$, $p = .82$). Based on AIC, no possible model incorporating a subset of these factors was superior in explaining the data than a simple intercept model (**Supplementary Table S4**). More importantly, none of the

models was superior to the previous model incorporating the electric field in the brain (Supplementary Table S3, S4).”

2. The effect size of tACS on MEG alpha power appears to be very small and to be driven by just two subjects. See Fig. 3b and Fig. 4c, d and e. Without the two subject showing alpha power increases close to 1 and 1.5e-24, the sham and the tACS groups would be unlikely to show any significant difference. Are these two subject’s outliers?

We thank the reviewer for this important observation. There are indeed 2-4 datapoints in the tACS group that show a relatively strong power increase compared to the rest of the sample. What is striking, however, is that this strong power increase is in fact predicted by our model (Fig. 5a), even if the to be predicted datapoint is not part of the training set (Fig 5c). These subjects appeared to be ones receiving optimal stimulation and therewith show the strongest power increase. To further support our results, we performed a replication experiment using a within subject design. We were able to replicate the effect. With the ability to contrast each subjects’ power increase after tACS against its individual sham control the effect seems relatively consistent (Fig 6).

Ln 182: “In order to investigate how much of participants’ individually sham controlled tACS effect the model can explain, and in order to replicate the previous results, we repeated the experiment using a within-subject design on a sample of 19 subjects. On two separate days, participants received tACS or sham stimulation for 20-min. The order of tACS and sham conditions were counterbalanced across participants.

As in the first experiment, a dependent samples random permutation cluster t-test revealed a stronger increase of participants’ source projected alpha power after tACS as compared to sham ($p_{cluster} < .001$, Fig. 6a-d). Source projected alpha power significantly increased after tACS ($p_{cluster} < .001$), and showed a trend towards increased alpha power after sham stimulation ($p_{cluster} = .08$). Again, there was no significant difference in the neighboring beta ($p_{cluster} > .16$) and theta frequency range ($p_{cluster} > .08$). Subsequently we tested whether participants’ individual stimulation effect in the alpha band (power increase after tACS – power increase after sham) within the significant cluster (Fig. 6a) are explained by our measures of tACS targeting. To this end, the power increase relative to sham was submitted to a multiple linear regression with factors $PRECISION_{Spat}$, $PRECISION_{Freq}$ and $STRENGTH$. Although only reaching trend-level, the model again explains a striking amount of the variance of the tACS effect ($R^2 = .62$, $F_{7,11} = 2.66$, $p = .07$, Fig. 5d, Fig. 6e-h). Specifically, $PRECISION_{Spat}$ significantly predicted participants’ individual stimulation effect ($\beta = 1.073e-24$, $t_{11} = 3.07$, $p = .01$). In addition, there was a trend towards an interaction between $PRECISION_{Spat}$, $PRECISION_{Freq}$, and $STRENGTH$ ($\beta = -3.6e-23$, $t_{11} = -1.75$, $p = .1$). Although the model performs weaker on the newly recorded dataset, results are generally in agreement with the findings of the previous experiment.”

3. In general results appear to be cloaked behind complex processing and statistics. This makes it difficult to judge the quality of the data and the size of the effect. The main effect of tACS on MEG alpha power is reported in section 2.2, where the authors use a random permutation cluster t-test on the source-projected alpha power. Why was this test used? The authors should report the results of a simple, standard, non-parametric test, such as

Wilcoxon sum-rank and signed-rank test, on the MEG alpha power with appropriate simple filtering (is source-projection necessary to detect an effect?).

We thank the reviewer for bringing up this important concern. However, we would like to emphasize that in the context of MEG we have to deal with highly complex data (306 channels), such that performing a single test on one sensor or a single source location is problematic.

For that reason, processing and statistics as employed in our experiment are commonly used in MEG research. The main motivation for testing effects on the source-level is to account for inter-individual differences in head-position and head shape while obtaining richer information about the source of observed effects. In contrast to the EEG, where electrodes are placed above standard electrode positions and sample signals from approximately the same regions, in the MEG participants place their heads in a fixed MEG helmet, allowing them to freely position their heads relative to the sensors. In addition, the sensor positions cannot be adapted to the individual head shape and size. This can give rise to differences across subjects which sensors pick up which activity. (E.g. the signals originating from occipital brain areas of a subject with a large head placed in the back of the helmet may be picked up by very posterior sensors, while signals from the same region of a subject with a small head placed towards the front of the helmet may be picked up by more central sensors). This renders comparisons over single channels difficult as additional, unpredictable variance can be introduced, which can lead to false positive or false negative results.

Projecting the data into source space allows to compare signals within the same brain regions thereby accounting for different head positions, shapes and sizes. In addition, source projection offers better interpretability of results as the activity can be linked to the relevant brain regions more directly.

The random permutation cluster t-test is an approach to analyze large scale datasets as encountered in the MEG and deals with multiple comparisons. The test computes the t-statistic for each datapoint (in our case source locations) and computes p-values in a non-parametric fashion by comparing the test data against random permutations. Subsequently, p-values are corrected for multiple comparisons. In contrast to applying a single/simple test on a single, pre-defined sensor or source location, the test can be performed without prior assumptions about the location of the effect. We therefore believe that our approach is more appropriate given the complexity of our data.

It is also not clear why MEG alpha power increase in both the tACS and sham group.

We thank the reviewer for pointing towards this missing information. We have devoted a new section in the discussion to the power increase in the sham group. In fact, an increase in alpha power has been reported in several studies, e.g. with time on task and has to be expected especially during monotonous experiments:

Ln 301: "In addition to the widespread power increase after tACS, we also observed an increase of power in the alpha band after sham stimulation that was not explained by our

statistical model. Such increase in alpha power is commonly observed with time-on-task and has been associated with vigilance decrement and mental fatigue^{39,50–54}”

The same applies for the use of clustering analysis. Why are only alpha power changes within specific clusters compared? What would happen to the effect if alpha power changes from the whole brain were included? The clusters used in the sham and tACS groups appear to be very different. Wouldn't a fairer comparison be to take the same brain regions in each group? What happens to the effect on alpha power if this approach is used? It's often unclear how and where the clustering analysis was used. Specifically, in lines 113 to 120, alpha power increase was extracted from specific clusters, were the PRECISIONfreq, PRECISIONspat and STRENGTH extracted from the same clusters?

We thank the reviewer for raising this point. Deciding about an appropriate region of interest (or clustering) that offers the fairest comparison between experimental groups was in fact the most difficult decision when conceiving the analysis pipeline. Using the significant cluster of the group comparison would have been a double dipping procedure as we are testing for group differences in the subsequent regression modelling. Pooling the data to test for the region with significant power increase relative to baseline to obtain a common region of interest might have introduced a bias in favor of the tACS group. The power increase is substantially larger and more wide-spread in this group and the effect in the pooled comparison would have been likely to be driven by the tACS group. Therefore we decided to use the group specific clusters as we think this offers the fairest comparison as it compares the significant power increase relative to baseline in both groups against each other and rather introduces a bias in favor of the sham group as we are only including data from the regions with the strongest power increase.

With respect to the second part of the question, the predictors cannot be extracted from the clusters, but were computed as described in the methods section. The STRENGTH parameter is computed by selecting the 10,000 strongest electric field magnitudes inside gray and white matter compartments and averaging them (ln 508). The PRECISIONfreq parameter is the stimulation frequency minus the peak frequency in the average spectrum over all MEG sensors during the baseline block (ln 515). The PRECISIONspat parameter is the spatial correlation between the simulated electric field and the topography of participants' alpha power source projected onto the individual brain (ln 495).

4. If the authors are confident in their models prediction power it should be properly validated with a second set of independent experiments. Take a new group of subjects, now use the model designed from the first set of experiments to predict how much each new subject's alpha power will increase. Then measure the actual alpha power change for each new subject and show that the model (based only on data from the first group) correctly predicts alpha power changes.

We thank the reviewer for this great suggestion. We collected a new dataset using a within subject design in order to validate/replicate our results. We tested if the model we identified previously can explain the individually sham-controlled tACS effect (power increase relative to baseline after tACS – power increase relative to baseline after sham). Although overall

only reaching trend level, the model again explains a large proportion of the variance ($R^2 = .62$, $F_{7,11} = 2.66$, $p = .07$, **Fig. 5d**, **Fig. 6e-h**).

It was, however, not possible to predict the power increase after tACS based on the model of the old experiment. This might, however, be in part explained by differing measurement conditions as compared to the previous experiment. Between the two experiments, the MEG system was disassembled and rebuilt which may have created differences in noise-levels. The recording time slots, although kept constant across individual measurements, were different as compared to the previous experiment. Further, the model is of course trained on a relatively small dataset that might not generalize well to the population. Nevertheless, the fact that we were able to replicate our previous results increases our overall confidence in the model.

Ln 182: “In order to investigate how much of participants’ individually sham controlled tACS effect the model can explain, and in order to replicate the previous results, we repeated the experiment using a within-subject design on a sample of 19 subjects. On two separate days, participants received tACS or sham stimulation for 20-min. The order of tACS and sham conditions were counterbalanced across participants.

Similar to the first experiment, a dependent samples random permutation cluster t-test revealed a stronger increase of participants’ source projected alpha power after tACS as compared to sham ($p_{cluster} < .001$, **Fig. 6a-d**). Source projected alpha power significantly increased after tACS ($p_{cluster} < .001$), and showed a trend towards increased alpha power after sham stimulation ($p_{cluster} = .08$). Again, there was no significant difference in the neighboring beta ($p_{cluster} > .16$) and theta frequency range ($p_{cluster} > .08$). Subsequently we tested whether participants’ individual stimulation effect in the alpha band (power increase after tACS – power increase after sham) within the significant cluster (**Fig. 6a**) are explained by our measures of tACS targeting. To this end, the power increase relative to sham was submitted to a multiple linear regression with factors $PRECISION_{Spat}$, $PRECISION_{Freq}$ and $STRENGTH$. Although only reaching trend-level, the model again explains a striking amount of the variance of the tACS effect ($R^2 = .62$, $F_{7,11} = 2.66$, $p = .07$, **Fig. 5d**, **Fig. 6e-h**).

Specifically, $PRECISION_{Spat}$ significantly predicted participants’ individual stimulation effect ($\beta = 1.073e-24$, $t_{11} = 3.07$, $p = .01$). In addition, there was a trend towards an interaction between $PRECISION_{Spat}$, $PRECISION_{Freq}$, and $STRENGTH$ ($\beta = -3.6e-23$, $t_{11} = -1.75$, $p = .1$). Although the model performs weaker on the newly recorded dataset, results are generally in agreement with the findings of the previous experiment. “

Further critical validations that could easily be carried out are missing:

a) If the tACS frequency – alpha frequency mismatch is important, then simply systematically change this parameter in an experimental setting for a number of subjects and show a relationship between frequency mismatch and alpha power.

b) If E-field amplitudes is important, then systematically change this parameter (stimulation at 0.5, 1 and 1.5 mA) and show a relationship between E-field amplitude and alpha power.

c) If overlap with the alpha topography is important, then change electrode configurations and show a clear change in alpha power.

Without these extra validations, the model/statistical prediction as presented is not completely convincing.

We thank the reviewer for the suggested validations. However, we feel that these experimental manipulations do not serve the purpose of our study. The goal of our study was to reveal the impact of randomly occurring inter-individual differences of electric fields and frequency mismatch when using the same stimulation parameters on tACS variability. Experimentally manipulating montages, intensities and frequency mismatches would result in a mixture of variance due to the individual and variance due to experimental variation that are difficult to disentangle. Given that each of the manipulations would require the collection of an additional sample of ~20 participants (60 recordings in total) we refrained from implementing these conditions.

5. The authors used ROAST for the segmentations of individual MRI data. ROAST is known to perform sub-optimally (compared to manual segmentation) in CSF and bone segmentation [1]. Both CSF and bone thickness will significantly contribute to the electric field strength and distribution in the brain. Therefore, given that we know this type of automated segmentation will introduce variability in individual head models, it is somewhat strange and puzzling that the authors find these head models can explain variability in the experimental data. This should be explained.

[1] Huang, Y., Datta, A., Bikson, M. & Parra, L. C. Realistic vOlumetric-Approach to Simulate Transcranial Electric Stimulation – ROAST – a fully automated open-source pipeline. bioRxiv 217331 (2019). doi:10.1101/217331

The reviewer raises a very important concern here. In fact, many steps in the simulation pipeline for the electric field modelling can influence the result (e.g. the conductivities chosen for the different compartments). After all, the model can only provide an estimate of the electric field in the brain that has some degree of uncertainty/prediction error. It is important to emphasize that a perfectly accurate prediction of the electric field is actually not required for our models to be predictive as long as the individual differences between participants are captured and the variance introduced by segmentation errors is random and not systematically biased such that it is artificially predictive of the effect.

We have refined the paragraph in the discussion section based on the reviewer's comment to further emphasize the role of uncertainty in the electric field predictions.

Ln 305: "As with all scientific studies, some limitations of the current findings deserve consideration. The individual electric fields used for our analysis were obtained from computational modelling. This approach can only provide predictions of the individual electric field with an inherent degree of uncertainty and simplification. For example, errors in the automatic tissue segmentation can add random error to the estimated field strengths. Recently, first efforts have been carried out to validate and calibrate results of current flow predictions using in-vivo electrophysiology^{31,35,36}. Results of these studies suggest that the models perform very well in predicting the spatial distribution of the induced electric fields, while tending to overestimate their strength. For our analysis approach, an accurate prediction of the exact field strength is not necessarily required, as long as the relative

difference in the fields across subjects is accurately represented and uncertainties in the estimates, e.g. due to segmentation errors, introduce error variance but no systematic bias. Noteworthy, conductivities used for simulations of the ROAST toolbox have recently been calibrated to increase the accuracy of voltage and field strength predictions³⁵. Our results indicate that both, the spatial distribution and the field strength predicted by individual electric field models, contain meaningful information allowing to predict the impact of tACS aftereffects, indicating that the computational models are sufficiently accurate to capture inter-individual differences. Nevertheless, further validation and optimization of electric field modelling using empirical data will be necessary to increase confidence in their predictions. Especially when models are integrated in the analysis of physiological or behavioral outcome measures as we propose in the current study, the accuracy of the utilized computational model will be crucial.“

Minor Comments

There appears to be a word missing from the first sentence in the abstract. I would suggest that ‘reducing variability’ or ‘understanding variability’ is the challenge. Every measurement or intervention has variability.

We thank the reviewer for spotting this error and corrected the sentence accordingly.

Introduction, line 47. ‘fail to replicate’ something missing in this sentence

We have rephrased the sentence to: “In recent years, tES methods received considerable criticism, arguing that stimulation effects are weak, highly variable or cannot be replicated¹⁸⁻²¹.”

Line 62-63 ‘the spatial correlation of the target brain activity (source of the α -oscillation) with the individually simulated electric field’ It’s not clear what this means

We refined the sentence for clarification: “Specifically, we tested whether the spatial correlation of the target brain activity (spatial pattern of the source projected α -oscillation) with the individually simulated electric field as well as the maximum field strength inside gray and white matter compartments can predict the variability of the power increase in the α -band after tACS.”

Line 66-67 ‘The former can be interpreted as a measure of precision (how well does the electric field match the pattern of the targeted brain activity),’ It’s not clear what this means. Which pattern is being referred to? Anatomy? Neural oscillation? It’s also not completely clear what the ‘former’ and ‘latter’ refer to here.

We thank the reviewer for pointing out this unclear formulation. We changed the wording to be more precise: “The spatial correlation provides a measure of precision, namely how well does the electric field match the spatial pattern of the targeted brain activity, which is the source of the α -oscillation in the current study. The maximum field strength provides a measure of the intensity at which the target activity can be perturbed.”

Line 74-75. The Introduction could be rounded off with a short statement of how the data/results support (or refute) the hypothesis stated in the last sentence.

We thank the reviewer for this suggestion. We added the following sentence to the end of the introduction section:

Ln 80: “Our results indicate that a complex interplay between the spatial precision and strength of the electric field along with the mismatch of the stimulation frequency and participants’ individual alpha frequency account for a large proportion of the variability of tACS effects in humans.”

Figure 1. What does ‘activation’ refer to on the correlation plot at the bottom of the figure? Units?

We changed the axis label to source power, which corresponds to the metric used in our analysis.

Line 90-91. ‘inter individual differences across subjects’ Should be either ‘inter individual differences’ or ‘differences across subjects’

We have changed the sentence to:

“Although we administered (and simulated) tACS with a fixed intensity of 1 mA (peak-to-peak) and the same Cz-Oz electrode montage (**Fig. 1c**), simulations of electric fields revealed differences across subjects in terms ...”

Figure 2. It’s good to show all the individual data. But maybe some could be moved to supplemental and the authors could pick out a few examples to show on a larger scale. MNI abbreviation in the figure legend is not clear.

We have implemented the requested changes to the figure. We are now showing larger versions of the electric fields of the first 10 subjects and moved the original figure to the Supplementary materials. MNI is now spelled out at the first instance.

Line 97. MNI abbreviation is not spelled out when it is first used.

MNI is now spelled out at the first instance in Ln 104.

Line 97-100. It’s not clear (at least at this point in reading the manuscript) why looking at the spatial correlation between different individuals is interesting or relevant.

We rephrased the sentence to motivate the reason for computing spatial correlations of the electric fields between participants.

Ln 103: “To characterize the similarity of electric fields across subjects, individual simulation results were warped into Montreal Neurological Institute (MNI)-space. Spatial correlations of the fields were computed between all subjects to attain insights into the overall variability of the factor.”

Line 104. *sample's* -> *samples*

We thank the reviewer for spotting this error. We corrected it in the revised manuscript

Figure 3. *Why is the power shown in arbitrary units?*

Due to the projection into source space there is no unit for the power. However, as the spectra are shown for the gradiometer sensors, we added the correct units there.

(Line 118, Line 132-133) *If this is the prediction, then why include the sham group in the model? Only model tACS group?*

The idea behind modelling the whole sample was to test if the prediction differs between groups (which it did) to rule out that the model captures some general mechanism related to the alpha power increase irrespective of stimulation.

Line 116 – *not clear what these clusters are?*

We added more detailed descriptions of the clusters In 123:

In order to evaluate whether the observed inter-individual differences of electric fields account for the variability of our outcome measure, for each subject, the average power increase between pre- and post-stimulation was extracted from the two group specific clusters, that is the cluster of each group exhibiting significant power increase from the pre- to the post-stimulation block **Fig. 4a,f**. The results were submitted to a multiple linear regression analysis with factors *CONDITION* (tACS vs. sham), *PRECISION_{spat}* (spatial correlation of α -topography with electric field), *PRECISION_{Freq}* (mismatch between stimulation frequency and individual α -frequency during the baseline block) and *STRENGTH* (average over 10,000 highest electric field magnitudes inside gray and white matter).

Reviewer #3 (Remarks to the Author):

Kasten et al., tested whether the inter-individual variability in the after-effects of 1 mA tACS in Oz-Cz montage at individual alpha frequency (IAF) on spontaneous posterior alpha power (i.e., a power increase in the alpha band) can be explained by several factors regarding the individual spatial and frequency specificity of the stimulation: i.e., the overlap of individual cerebral stimulation current distribution (estimated from anatomically informed electric field simulations) and topography of the targeted alpha oscillatory power (determined from resting state MEG source localization), the maximal current strength in the brain (from the simulations), and the divergence of the stimulation frequency (determined from a 3 min resting MEG session prior to the experiment) from the actual individual alpha frequency (determined during an MEG baseline block). They compared their experimental group (N = 20) to a control group (N = 20) that received conventional sham stimulation only in a between subject-design. The authors report remarkably high R² coefficients of 87 % (original full model) to 52 % (leave one out cross-validation) explained variance. This is a highly interesting paper. It is clearly written and the data processing and statistical

analyses are sound. The conclusions are principally backed up by the results, although there are some limitations (see below). If these exciting results hold true, this paper would set the ground for establishing individualized stimulation protocols that allow the induction of after-effects (and presumably also immediate online effects, such as neuronal entrainment) with conventional low currents (here 1 mA peak-to-peak). This is in line with several recent studies showing considerable effects of low-intensity tACS on cognition (such as working memory or memory encoding) when precisely targeting networks in an individually tailored fashion. This paper is also timely and of particular importance given recent studies questioning the effectiveness of low-intensity tACS altogether. However, given the relevance and potential implications of the reported findings, I would encourage the authors to further corroborate their conclusions by implementing at least some of the control experiments suggested below. I have the following specific comments:

1. The decision for a between-subject design over a within-subject design is unfortunate (in particular because of the huge inter-individual variability which is in the focus of this paper), as it prevents the prediction of individual sham-controlled tACS after-effects. Instead the authors had to employ a general linear model including the stimulation (tACS vs sham) as factor and the spatial and frequency specificity as covariates (or a multiple linear regression analysis, as the authors put it, with the stimulation as between-subject dummy variable, and interactions with that factor as relevant comparisons. Luckily, however, the predictors also explained substantial variance for the tACS group alone but not for the sham group, however in that case obviously without a direct statistical tACS-sham comparison and with a reduced sample size of 20 subjects, which is at the edge for a proper correlation analysis. Also, the leave-one-out cross-correlation is not a very strong validation check, and yet it was associated with a considerable drop from 87 % to 53 % explained variance. It would therefore be much more convincing if the results would be replicated with a new independent sample, preferably in a within-group design.

We thank the reviewer for bringing up this very important aspect. We performed the suggested replication experiment using a within subject design on 19 participants. Although the overall model only reached a trend, it still explains 62% of the variance of the individually sham controlled tACS effect on participants' source-level alpha power ($R^2 = .62$, $F_{7,11} = 2.66$, $p = .07$, **Fig. 5d, Fig. 6e-h**), supporting the results of our previous experiment.

Ln 181: In order to investigate how much of participants' individually sham controlled tACS effect the model can explain, and in order to replicate the previous results, we repeated the experiment using a within-subject design on a sample of 19 subjects. On two separate days, participants received tACS or sham stimulation for 20-min. The order of tACS and sham conditions were counterbalanced across participants.

Similar to the first experiment, a dependent samples random permutation cluster t-test revealed a stronger increase of participants' source projected α -power after tACS as compared to sham ($p_{cluster} < .001$, **Fig. 6a-d**). Source projected α -power significantly increased after tACS ($p_{cluster} < .001$), and showed a trend towards increased α -power after sham stimulation ($p_{cluster} = .08$). Again, there was no significant difference in the neighboring β - ($p_{cluster} > .16$) and θ -frequency range ($p_{cluster} > .08$). Subsequently we tested whether participants' individual stimulation effect in the α -band (power increase after tACS – power increase after sham) within the significant cluster (**Fig. 6a**) are explained by our measures of tACS targeting. To this end, the power increase relative to sham was submitted to a multiple

linear regression with factors $PRECISION_{Spat}$, $PRECISION_{Freq}$ and $STRENGTH$. Although only reaching trend-level, the model again explains a striking amount of the variance of the tACS effect ($R^2 = .62$, $F_{7,11} = 2.66$, $p = .07$; **Fig. 5d, Fig. 6e-h**). Specifically, $PRECISION_{Spat}$ significantly predicted participants' individual stimulation effect ($\beta = 1.073e-24$, $t_{11} = 3.07$, $p = .01$). In addition, there was a trend towards an interaction between $PRECISION_{Spat}$, $PRECISION_{Freq}$, and $STRENGTH$ ($\beta = -3.6e-23$, $t_{11} = -1.75$, $p = .1$). Although the model performs weaker on the newly recorded dataset, results are generally in agreement with the findings of the previous experiment.

2. The most convincing evidence would be, however, if the authors would specifically optimize the current flow in a given subject (based on its electric field simulations) using a multi-electrode setup for targeting its individual alpha power distribution and compare it to the standard Cz-Oz montage. If the optimized setup produces much larger after-effects, this would be a very strong argument.

We thank the reviewer for this suggestion. Indeed, a framework/protocol that causes reliable stimulation effects by individually optimizes stimulation parameters to best match the individual subject would be very desirable and have huge impact on the brain stimulation community. However, at the same time the implementation of such a framework is not straight forward and requires intensive development and testing of the different modules involved (1: an algorithm that is able to propose a stimulation montage based on a source-localization as input; 2: A more reliable method to determine the dominant frequency in a given frequency band. 3: Ideally this method would adapt to systematic drifts over time, which requires a closed-loop system with online artifact correction). We therefore considered the requested experiment to not be realizable within a reasonable time-frame as part of this study and decided to perform the previously mentioned replication experiment instead.

3. As the authors discuss, recent evidence published in this journal (Asamoah et al., 2019) has demonstrated the impact of transcutaneous stimulation effects, and also retinal stimulation has been highlighted as potential confound (Schutter, 2016). The authors argument against such an influence explaining their effects is reasonable, i.e., that the predictive value of the overlap between current distribution and target oscillation topography in the brain cannot be explained easily with such sensory confounds. However, the predictive value of the current density in the skin and eye balls has not been investigated, and there is no other control condition, such as a different stimulation frequency or different electrode positions. Is the observed alpha power increase thus really due to synaptic changes following alpha entrainment? Or would another frequency or stimulation target have resulted in similar predictions? Maybe the spatial correlation between current distribution and alpha power distribution is mainly driven by individual differences in the alpha power topography? A frequency control (stimulating at flanker frequencies) or montage control (stimulating in a different electrode configuration) may thus strengthen the conclusions.

Based on the reviewers' excellent suggestion we implemented an additional control analysis. We extracted the maximum current strength inside the eyeballs and the skin to create an alternative statistical model aiming to predict the power increase after tACS based on peripheral mechanisms. In contrast to the electric field inside the brain, this model failed to

significantly predict our data and was not superior to a simple intercept model. We added the new results to the manuscript starting in Ln 158:

“Recently, concerns have been raised that tACS effects may not originate from electric stimulation of the brain, but exhibit its effects indirectly via stimulation of peripheral nerves (e.g. stimulation of the retina or transcutaneous nerves)^{42,43}. Our results indicate that the extent of the tACS aftereffect can be predicted using the electric field inside the brain, which is difficult to explain with such peripheral mechanisms of action. We therefore conducted an additional analysis aiming to explain the data in our tACS group by a model incorporating the maximum current in the skin (*STRENGTH_{skin}*; average over the maximum 10,000 voxels within the skin compartments) and the eyeballs (*STRENGTH_{eye}*; average over the maximum 1000 voxels within the eyeballs). In addition, we included the factor *PRECISION_{Freq}* from our initial model as a similar effect of frequency mismatch has to be expected for peripheral stimulation effects. The resulting model was not able to significantly predict the power increase after tACS ($R^2 = .22$, $F_{7,12} = 0.49$, $p = .82$). Based on AIC, no possible model incorporating a subset of these factors was superior to a simple intercept model in explaining the data (**Supplementary Table S4**). More importantly, none of the models was superior to the previous model incorporating the electric field in the brain (**Supplementary Table S3, S4**).”

4. Page 6: The authors tested for frequency specificity only with respect to the EEG power spectrum, and did not find neighboring theta and beta range to be affected. However, with $p = 0.18$ and $p = 0.19$, the beta error (false negative) due to low power is presumably not very low... and may have been statistically significant with a larger N?

We repeated the analysis on the neighboring frequency bands in our replication experiment that should have slightly larger statistical power due to the within subject design. Again, we find rather small, yet insignificant p-values (.16 and .08). It thus cannot be ruled out that neighboring frequencies may have been affected by the stimulation. It can, however, be argued that the effect is likely to be a lot smaller than the effect in the alpha band. Nevertheless, we would like to emphasize that it was not the primary goal of the study to provide evidence in favor or against the frequency specificity of tACS aftereffects, but to study the influence of individual differences on the tACS aftereffect in the range of the stimulation frequency.

5. Page 6: comparing cluster extent after DICS beamforming is problematic, even after a significant cluster permutation test, since differences in cluster extent are not independent from differences in oscillatory power. Is there really a broader activation or just a stronger activation of the same network (and would lowering the threshold reveal the same regions in the sham condition)?

We thank the reviewer for pointing at this important issue. We have added a note about the dependence between cluster extent and power enhancement to the discussion.

Ln 291: “To our surprise, the effect of tACS in the α -band was very widespread covering a large proportion of the cortex, including frontal areas not covered by our electrode montage. We did not further investigate this observation up to this point as it was beyond the scope of our main research question. However, there is evidence that distributed brain

networks communicate via correlated activity within specific frequency bands⁵¹. It might thus also be possible that the tACS-induced modulation of oscillatory activity within a circumscribed region could lead to co-stimulation of distant brain areas functionally coupled via the stimulated frequency band. It should however also be emphasized that differences in cluster extent are not independent of oscillatory power and might thus be solely explained by the power enhancement in the alpha band.”

6. Also: Is there a direct link between the voxel-wise stimulation current strength and the subsequent increase in alpha power in that voxel?

We computed the voxel-wise stimulation current strength with the individually sham controlled tACS effect in our second experiment (Supplementary Fig. S2). Even without correction for multiple comparisons there are only very few regions showing a significant correlation between tACS effect and the electric field strength in that voxel.

7. Caption Figure 4: “The cluster from a group analysis was used to extract individual alpha power increase values of each subject in the tACS group for subsequent regression analysis” Using a group ROI for extracting that information introduces a bias in a way that it may result in larger values for subjects with an alpha topography more similar to the group average (and thus with a larger overlap of their individual cluster and the group average cluster). The approach is understandable. But what happens if using the x largest values from voxels of the individual alpha power cluster for extraction? The same holds true for extracting the maximum current strength from the simulations. Why not try to extract it only for the relevant voxels, i.e., the ones overlapping with the (individual?) alpha power cluster?

We thank the reviewer for raising this point. We repeated the analysis as suggested for the data of the first experiment and obtained very similar results for our models. The results have been added to the Supplementary Materials:

“Using a group ROI can introduce a bias such that larger power values are observed for participants whose alpha power distribution on the source level is more similar to the cluster. We thus repeated our analysis identifying and averaging over the 1000 source locations within the clusters that show the strongest alpha power increase to baseline for each participant. We submitted these power values to our linear regression analysis with factors CONDITION, PRECISION_{Freq}, PRECISION_{Spat}, and STRENGTH. We obtained similar results as for our ROI analysis in section 2.3. The model significantly predicted participants peak power increase in the ROI ($R^2 = .78$, $F_{15,24} = 5.89$, $p < .001$). Specifically, the factors CONDITION ($\beta = 7.203e-25$, $t_{24} = 3.33$, $p = .003$), the interaction between CONDITION, PRECISION_{Freq} and STRENGTH ($\beta = 5.114e-23$, $t_{24} = 3.24$, $p = .003$) and the interaction between CONDITION, PRECISION_{Freq}, PRECISION_{Spat} and STRENGTH ($\beta = 2.896e-22$, $t_{24} = 4.07$, $p < .001$) significantly predicted the power increase. When separately fitted to the data of the two groups, the model again failed to explain the power increase in the sham group ($R^2 = .15$, $F_{7,12} = 0.31$, $p = .93$), but significantly predicts the power increase after tACS ($R^2 = .82$, $F_{7,12} = 7.58$, $p = .001$). Specifically, the factors PRECISION_{Spat} ($\beta = 3.90e-24$, $t_{12} = 3.74$, $p = .003$), the interactions between PRECISION_{Spat} and PRECISION_{Freq} ($\beta = 4.41e-24$, $t_{12} = 2.73$, $p = .018$), STRENGTH and PRECISION_{Freq} ($\beta = 4.53e-23$, $t_{12} = 4.89$, $p < .001$) and PRECISION_{Freq}, PRECISION_{Spat} and STRENGTH ($\beta = 2.90e-22$, $t_{12} = 6.56$, $p < .001$) significantly predicted participants peak power increase after tACS. “

With regard to the second part of the question, we did try to extract the current from the voxels showing maximum alpha power. However, we recognized that this predictor is not statistically independent from the spatial correlation, as higher similarity of the electric field and the source activation entails stronger electric fields at the voxels of maximum alpha power. Therefore, we decided to use the maximum strength of the electric field regardless of the region as an independent predictor, while still being able to disentangle the role of precision and the role of the current strength. Our results indicate that both play a role. However, it could have been possible that, for example, the current strength arriving inside the cortex is relatively similar across subjects while the spatial correlation explains most of the variability or vice versa. In such a case one of the predictors would have not survived our model selection procedure.

8. It is surprising that the deviation of the tACS frequency from the IAF was a relevant predictor, given that tACS frequency had deliberately been adjusted to the IAF beforehand. How can there still be a meaningful divergence? Is this explained by a substantial change in IAF from resting MEG to baseline MEG session? What is the actual resolution of calculating the IAF for each measurement and what is the resolution of the actual stimulation frequency (1 Hz, 0.5 Hz, 0.25 Hz?). The actual IAF values and stimulation frequency values should be reported in detail and the issue should be discussed.

We thank the reviewer for this important point, that has not been sufficiently discussed in the first version of the manuscript. There is evidence that the frequency of the alpha oscillation drifts over time and during different background tasks which can lead to discrepancies between the dominant frequency identified prior to the experiment and during the baseline block. In some cases, we also observed two peaks in the alpha range, of which we choose the larger one as the stimulation frequency. Subsequently, the discarded peak may have turned to the dominant one. Due to technical reasons (the way our signal generator is controlled), we also have to round our stimulation frequency to the next integer frequency. The IAF was determined with a resolution of 0.5 Hz. The analysis of the baseline block was performed with a frequency resolution of 0.25 Hz. Together, these factors can create a mismatch between the tACS frequency and the dominant frequency in the experiment, that apparently contributes to the variability of stimulation effects. We added some sections to the manuscript detailing the issue:

Ln 74: While the frequency of the alpha oscillation has long been assumed to be relatively stable, more recent evidence suggests that alpha frequency can exhibit substantial intra-individual variability across different tasks and over time^{39,40}.

Ln 274: “Despite tuning tACS to each participants’ individual alpha frequency as measured prior to the experiment, there was still a mismatch between the stimulation frequency and the individual alpha frequency observed during the experiment that significantly contributed to the variability of the power increase after tACS. This mismatch has previously been reported to occur despite applying stimulation at participants’ individual frequency and to affect the extend tACS aftereffects^{17,38}. Different processes may explain the occurrence of a frequency mismatch between tACS and brain oscillations. Firstly, the dominant frequency in a specific band may undergo changes over time. For example, systematic drifts of the individual alpha frequency have been observed over time and depending on the background

task^{39,40}. Secondly, for practical reasons in the current study the identified power peak in the alpha band was rounded to the next integer frequency which naturally gives rise to mismatches between stimulation frequency and the frequency of the targeted brain oscillation. Given the impact of this factor future studies might benefit from improved procedures to estimate tACS frequency.”

Ln 383: The power peak in the α -band between 8 Hz and 12 Hz within a fixed set of posterior gradiometer sensors (**Supplementary Table S5**) was identified and the closest integer frequency to the identified peak was used as stimulation frequency during the following experiment.

In addition, we added a table showing stimulation frequency, IAF determined prior to the experiment, the IAF during the baseline block of the experiment and the frequency mismatch between stimulation frequency and the IAF during the baseline block to the Supplementary Materials.

9. Only current strength but not current direction relative to the cortical surface is investigated. This factor should at least be discussed.

We thank the reviewer for this important suggestion. We devoted a brief paragraph in the discussion to this aspect:

Ln 326: “The predictions of the tACS aftereffect in the current study are based on predictors derived from the magnitude of the electric field inside the brain, ignoring the direction of the field relative to the cortical surface. While the current flow radial to the cortical surface (or normal component of the electric field) determines the strength of somatic polarization of cortical pyramidal cells, the current flow radial to the cortical surface polarizes horizontally arranged cortico-cortical axons⁵⁶. In principle, these different components of the electric field could differentially contribute to stimulation effects. Models could incorporate this contribution by computing spatial correlations with the brain activity of interest and the strength parameter for each of the field directions. In the current experiment, we refrained from applying such models to the data as we aimed to keep statistical models sufficiently simple and interpretable. “

10. Page 3: “While tDCS is thought to exhibit its effect by changing neuronal excitability via alterations of neuron’s resting membrane polarization, tACS is believed to work via the principles of neuronal entrainment.” How is this entrainment effect mediated? By rhythmic shifts in the membrane potential. The immediate mechanisms of action on the neuronal level should be the same for tDCS and tACS, but the consequences may be different.

We thank the reviewer for pointing this out. We agree that the section was not phrased ideally. We have changed it to:

“While tDCS is thought to exhibit its effect by changing neuronal excitability via tonic alterations of neuron’s resting membrane polarization^{1,8-10}, the rhythmic shifts in the membrane potentials during tACS are believed to result in neuronal entrainment^{2,11}.”

11. Please elaborate on the tSSS method. (page 18)

We have added a more detailed description of the method in Ln 442:

The tSSS method decomposes the MEG signal into spatiotemporal components originating from inside and outside the helmet. The method is commonly used to suppress external artifacts and interference signals, especially those originating in the proximity of the head (e.g. implants, deep-brain stimulator, etc.)^{66,67,69}. The method is thus well suited to remove interference brought into the MEG helmet via the cables connected to the stimulation electrodes. Further, it allows to compensate for head-movements by transforming the signals to the initial head position⁶⁸.

12. Page 19: please provide criteria for ICA component rejection as well as mean and SD for the rejected components.

We thank the reviewer for this suggestion. We added the requested information to the Methods section:

Ln 452: "After visual inspection of component topographies and time-courses an average of 3.7 (\pm SD: 1; min: 3, max: 8) components were removed before back-projecting the signals into sensor-space. In experiment 2 3.6 (\pm SD: 0.9; min: 2, max: 6) components were rejected on average. Rejection criteria were based on recommendations in the literature⁷⁰"

References

Asamoah B, Khatoun A, Mc Laughlin M (2019) tACS motor system effects can be caused by transcutaneous stimulation of peripheral nerves. *Nat Commun* 10:266.

Schutter DJ (2016) Cutaneous retinal activation and neural entrainment in transcranial alternating current stimulation: A systematic review. *Neuroimage* 140:83-88.

REVIEWERS' COMMENTS:

Reviewer #2 (Remarks to the Author):

I the authors have provided a detailed rebuttal to my original review. The additional experiments and analysis help answer many of my original concerns. However, given the controversy in the tACS field at the moment with some papers claiming that the electric field in the brain is too weak to have an effect (Lafon et al 2017, Vöröslakos et 2018) and other claiming that effects are caused by retinal (Schutter et al 2016) or cranial nerve stimulation (Asamoah et al 2019), my main concern that the effects of tACS on alpha may not be caused the electric field in the brain has not been fully addressed. While, I commend the authors for collecting an additional independent dataset on which to test the statistical model, the model only predicts the new data at trend level and does not reach significance. I give more details on each of the my 5 original major comments. All the minor comments have been satisfactorily addressed.

Major Comments

1) The additional statistical analysis provided by the authors show that the electric field in the retina (phosphene stimulation) and the electrical field in the skin (peripheral nerve stimulation) do not predict tACS induced alpha power increase, while the electric field in the brain does. This does provide some support for the hypothesis that the observed increased in alpha power is caused by the field in the brain and not the field in the skin or eye. However, given that the original correlation was driven by a few points (see Major Comment 2) and the fact that the new data to validate the model do not reach significance level (see Major Comment 4), the prediction power of the model does not appear to be very strong. Given this, it's difficult to place too much weight on the fact that the model predicts that that skin and eye stimulation are not causing the increase in tACS alpha. I still believe the most convincing way to show this is through experimental evidence: thus using a protocol that has a focused montage to rule out retinal stimulation, scalp anesthetics to rule out peripheral nerve stimulation, or alternative montage locations that show even when skin and eye are stimulated, stimulating a a different part of the brain has no effect on alpha power. However, this would require collection of new experimental data.

2) The authors reply to this comment mainly focused on the new experimental data collected. This overlaps with Major Comment 4, and I address this there.

3) The authors have fully answered my concerns here. I would suggest to include some of the explanation about the MEG statistics into the Methods to make this clearer for other readers.

4) I greatly appreciate that the authors have taken the time to collect a second independent set of experimental data to help validate the model. However, while the statistical model developed on the first dataset does have some predictive power for the second dataset ($R^2 = 0.62$), it does not reach commonly accepted level of statistical significance ($p=0.07$). An alternative model, one which only includes spatial correlation, was needed to reach statistically significant predictive power ($p=0.01$). I assume that this is because the second dataset does not have the same large increases in alpha power observed in two subjects. I previously questioned in Major Comment 2 if these points were outliers. The new experiment seems to indicate that they were, and that without them the predictive power of the model is greatly decreased. Given that the main claim of the manuscript is that the model can be used to predict changes in alpha power caused by tACS, the supporting evidence provided by the second dataset is not very compelling, specifically because only a trend was observed and statistical significance was not reached.

5) The authors have answered this concern.

Minor Comments

Line 200-201. "Although the model performs weaker on the newly recorded dataset, results are generally in agreement with the findings of the previous experiment". Referring to this as "newly recorded" will not be logical for a reader of the final manuscript.

Reviewer #3 (Remarks to the Author):

The authors have sufficiently addressed all of by previous concerns and even conducted additional experiments and analyses: (a) a within-subject sham controlled replication in 19 new subjects, and (b) analyses of potentially confounding currents in eye balls and skin. Even though the results of replication experiment were slightly weaker, it nonetheless sorroborates the principal conclusion from the main experiment. I have no further comments and support publication of this paper in Nature Communications.

Response to Reviewer

REVIEWERS' COMMENTS:

Reviewer #2 (Remarks to the Author):

I the authors have provided a detailed rebuttal to my original review. The additional experiments and analysis help answer many of my original concerns. However, given the controversy in the tACS field at the moment with some papers claiming that the electric field in the brain is too weak to have an effect (Lafon et al 2017, Vöröslakos et 2018) and other claiming that effects are caused by retinal (Schutter et al 2016) or cranial nerve stimulation (Asamoah et al 2019), my main concern that the effects of tACS on alpha may not be caused the electric field in the brain has not been fully addressed. While, I commend the authors for collecting an additional independent dataset on which to test the statistical model, the model only predicts the new data at trend level and does not reach significance. I give more details on each of the my 5 original major comments. All the minor comments have been satisfactorily addressed.

Major Comments

1) The additional statistical analysis provided by the authors show that the electric field in the retina (phosphene stimulation) and the electrical field in the skin (peripheral nerve stimulation) do not predict tACS induced alpha power increase, while the electric field in the brain does. This does provide some support for the hypothesis that the observed increase in alpha power is caused by the field in the brain and not the field in the skin or eye. However, given that the original correlation was driven by a few points (see Major Comment 2) and the fact that the new data to validate the model do not reach significance level (see Major Comment 4), the prediction power of the model does not appear to be very strong. Given this, it's difficult to place too much weight on the fact that the model predicts that that skin and eye stimulation are not causing the increase in tACS alpha. I still believe the most convincing way to show this is through experimental evidence: thus using a protocol that has a focused montage to rule out retinal stimulation, scalp anesthetics to rule out peripheral nerve stimulation, or alternative montage locations that show even when skin and eye are stimulated, stimulating a different part of the brain has no effect on alpha power. However, this would require collection of new experimental data.

Authors: We thank the reviewer again for raising this important point. In accordance with the editor we believe that an additional experiment would be beyond the scope of the current study. However, we are taking the reviewers concern very seriously and discuss limitations of our control analysis in the discussion section along with suggestions for follow up experiments that could address these concerns and increase confidence in the use of electric field modeling to control for peripheral effects. For example, by explicitly showing that, if peripheral mechanisms are targeted (e.g. the retina rather than the brain) the models can be used to predict effects from peripheral mechanisms. In that case, the model could be used to disentangle peripheral stimulation effects from those originating inside the brain.

We added the following section to the discussion:

“Especially when models are integrated in the analysis of physiological or behavioral outcome measures as we propose in the current study, the accuracy of the utilized computational model will be crucial. This also holds true if the models are used to rule out peripheral stimulation effects. In the current study, we tested whether the current inside the skin or the eyeballs might be better suited to explain the observed power increase, which seemed not to be the case. This can be seen as evidence against an influence of such mechanisms on our effects. However, it is possible that the utilized computational models do not optimally predict the electric fields inside these compartments. To increase confidence in this kind of control analysis, future studies could assess whether the electric field modeling is able to accurately predict stimulation effects if peripheral mechanisms are explicitly targeted (e.g. by a montage optimized to stimulate the retina rather than the brain). Evidence for reliable separation of stimulation effects originating inside the brain and in the periphery would support the potential of electric field model to substitute additional control conditions currently required to rule out peripheral mechanisms to be underlying tACS effects. “

And explicitly emphasize that no focused montage or scalp anesthetics were used in the study:

“While our experiment did not rule out a possible impact of peripheral stimulation by applying scalp anesthetics or a focused stimulation montage, the aftereffect observed in our study seems very unlikely to be explained by predictors derived from the electric field inside the brain if such peripheral mechanisms had primarily caused the effect.”

2) The authors reply to this comment mainly focused on the new experimental data collected. This overlaps with Major Comment 4, and I address this there.

Authors: Accordingly, the reply can be found after Comment 4.

3) The authors have fully answered my concerns here. I would suggest to include some of the explanation about the MEG statistics into the Methods to make this clearer for other readers.

Authors: We thank the reviewer for this suggestion and have added a more detailed explanation of the MEG statistics to the DICS beamforming section:

“To test whether the power increase in the α -band was larger after tACS as compared to sham, power differences were submitted to a one-sided non-parametric random permutation cluster t-test with 10,000 randomizations and Monte Carlo estimates to calculate p -values. The approach allows to test for statistical differences in large scale datasets without the need for prior assumptions about the location of effects and while controlling for multiple comparisons. We decided to test for stimulation effects exclusively in the source-space in order to circumvent additional variance due to individual differences in head-size or position inside the MEG helmet which naturally limit the analysis of MEG sensor-space signals.”

4) I greatly appreciate that the authors have taken the time to collect a second independent set of experimental data to help validate the model. However, while the statistical model developed on the first dataset does have some predictive power for the second dataset ($R^2 = 0.62$), it does not reach commonly accepted level of statistical significance ($p=0.07$). An alternative model, one which only includes spatial correlation, was needed to reach statistically significant predictive power ($p=0.01$). I assume that this is because the second

dataset does not have the same large increases in alpha power observed in two subjects. I previously questioned in Major Comment 2 if these points were outliers. The new experiment seems to indicate that they were, and that without them the predictive power of the model is greatly decreased. Given that the main claim of the manuscript is that the model can be used to predict changes in alpha power caused by tACS, the supporting evidence provided by the second dataset is not very compelling, specifically because only a trend was observed and statistical significance was not reached.

Authors: The reviewers concern on the first model seems largely based on the question whether two of the datapoints in the experiment are outliers. In our first reply to this concern (comment 2) we argued that these datapoints are indeed extreme relative to the rest of the sample. However, as seen in our cross-validation analysis, these extreme values seem to be plausible (i.e. to be expected) within our model prediction (Fig. 5c), indicating that these subjects may have received tACS at more optimal stimulation parameters than the rest of the sample. Unfortunately, the reviewer did not comment on this argumentation). To further investigate the role of these two datapoints we added an additional analysis as Supplementary Note 1 and Supplementary Figure 3. In summary, we find that the two values can indeed be considered extreme as they exceed 2 SD from the mean. However, they do not seem to exclusively or excessively drive our model predictions, as even without them clear trends towards our main results are still evident. Further, using our model we can predict the power increase of these two subjects and find that given their electric field parameters (spatial correlation, strength and mismatch), their extreme power increase is in fact to be expected. We therefore do not think that it would be justified to remove these subjects from the analysis, or that they invalidate our conclusions.

We added the following segment to the Supplementary Information:

Supplementary Note 2 – Outlier analysis Experiment 1

Inspection of the data of experiment 1 may suggest the presence of extreme values/outliers in the data (Fig. 3b). Such extreme values can artificially drive the observed relationship between our predictors and the power increase after stimulation. Z-transformation of the power increase after stimulation within the group specific ROIs revealed two values exceeding two standard deviations from the mean of the whole sample ($Z_{\text{outlier1}} = 2.49$, $Z_{\text{outlier2}} = 4.34$). In order to test whether these two datapoints excessively drove the model fits, we repeated the regression analysis applied to the tACS group after removing the two datapoints. Without the two datapoints, the model still accounts for 62% of the variance (multiple linear model, $R^2 = .62$, $F_{7,10} = 2.34$, $p = .11$), although not reaching significance. The factor $\text{PRECISION}_{\text{spat}}$ (same multiple linear model, $\beta = 1.30\text{e-}24$, $t_{10} = 3.16$, $p = .01$) significantly predicted the data. Further, the model showed trends for the $\text{STRENGTH} * \text{PRECISION}_{\text{Freq}}$ (same multiple linear model, $\beta = 1.25\text{e-}23$, $t_{10} = 2.20$, $p = .052$) and the $\text{STRENGTH} * \text{PRECISION}_{\text{Freq}} * \text{PRECISION}_{\text{spat}}$ interactions (same multiple linear model, $\beta = 8.77\text{e-}23$, $t_{10} = 1.79$, $p = .10$). While the model performs weaker after removal of the two datapoints (which can at least in part be explained by reduced sensitivity of the model due to the smaller sample size), it still generally supports the direction of our findings. In a next step, we tested whether this model trained on the remaining datapoints would predict the occurrence of these outliers based on their electric field parameters and stimulation frequency mismatches. As depicted in Supplementary Figure 3, the model tends to underestimate the power increase after stimulation in these two datapoints. However, remarkably, the model accurately predicts both datapoints to be extreme values as compared to the rest of the sample. Taken together, we conclude that while the two datapoints

reflect extreme values as compared to the rest of the sample, they are in agreement with the model predictions. I.e. compared to the rest of the sample, the spatial extent of the electric field, its intensity and the stimulation frequency were most optimal and thus gave rise to the strong power increase that was observed after stimulation.

In addition, we inserted a brief discussion and reference to the Supplementary Note in the main manuscript:

Noteworthy, the tACS group contains two subjects that apparently exhibited exceptionally strong alpha power increases relative to baseline (Fig. 3b), which were very well predicted by the cross-validation model (Fig. 5c). Even a model that was trained without both of these subjects predicted the comparatively strong power increase based on their electric field parameters and frequency mismatches (Supplementary Note 1, Supplementary Figure 3).

5) The authors have answered this concern.

Minor Comments

Line 200-201. “Although the model performs weaker on the newly recorded dataset, results are generally in agreement with the findings of the previous experiment”. Referring to this as “newly recorded” will not be logical for a reader of the final manuscript.

We thank the reviewer for pointing this out. We rephrased the sentence to:

“Although the model performs weaker on the replication dataset, results are generally in agreement with the findings of the previous experiment.”

Reviewer #3 (Remarks to the Author):

The authors have sufficiently addressed all of my previous concerns and even conducted additional experiments and analyses: (a) a within-subject sham controlled replication in 19 new subjects, and (b) analyses of potentially confounding currents in eye balls and skin. Even though the results of replication experiment were slightly weaker, it nonetheless corroborates the principal conclusion from the main experiment. I have no further comments and support publication of this paper in Nature Communications.

Authors: We thank the reviewer for the helpful remarks on the manuscript and appreciate the positive evaluation of the work.